# LangTime: A Language-Guided Unified Model for Time Series Forecasting with Proximal Policy Optimization

Wenzhe Niu [* 1]   Zongxia Xie [* 1]   Yanru Sun [* 1]   Wei He [2]   Man Xu [3]   Chao Hao [1]

## Abstract

Recent research has shown an increasing interest in utilizing pre-trained large language models (LLMs) for a variety of time series applications. However, there are three main challenges when using LLMs as foundational models for time series forecasting: (1) Cross-domain generalization. (2) Cross-modality alignment. (3) Error accumulation in autoregressive frameworks. To address these challenges, we proposed **LangTime**, a **lan**guage-**g**uided unified model for **time** series forecasting that incorporates cross-domain pre-training with reinforcement learning-based fine-tuning. Specifically, LangTime constructs Temporal Comprehension Prompts (TCPs), which include dataset-wise and channel-wise instructions, to facilitate domain adaptation and condense time series into a single token, enabling LLMs to understand better and align temporal data. To improve autoregressive forecasting, we introduce TimePPO, a reinforcement learning-based fine-tuning algorithm. TimePPO mitigates error accumulation by leveraging a multidimensional rewards function tailored for time series and a repeat-based value estimation strategy. Extensive experiments demonstrate that LangTime achieves state-of-the-art cross-domain forecasting performance, while TimePPO fine-tuning effectively enhances the stability and accuracy of autoregressive forecasting.

## 1. Introduction

Time series refers to sequences of data points indexed in discrete-time order (Box et al., 2015), and they are common in real-world applications, such as financial risk as-

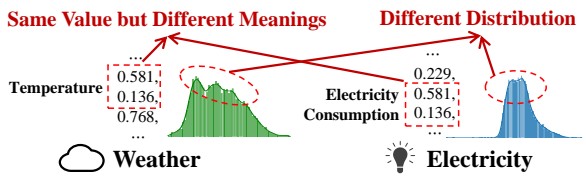

(a) Challenge 1: Cross-domain generalization.

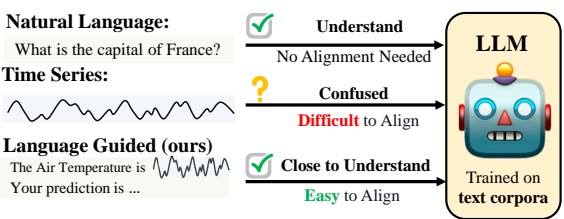

(b) Challenge 2: Cross-modality alignment.

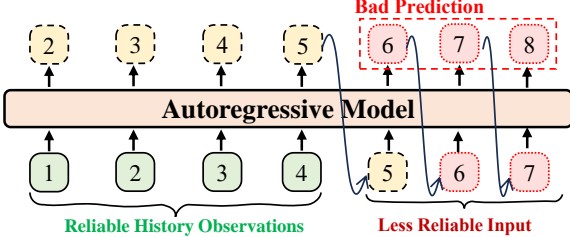

(c) Challenge 3: Error accumulation.

*Figure 1.* Challenges of Applying LLMs to Time Series. (a) Cross-domain generalization: Different domains possess their unique characteristics and numerical implications. (b) Cross-modality alignment: LLMs struggle to directly comprehend unseen time series as they are trained on language data, whereas language guidance can assist in enhancing their understanding. (c) Error accumulation: In autoregressive prediction, using the output from the previous step as input is considered unreliable.

sessment (Baffour et al., 2019), energy sustainability (Yu et al., 2023b), and weather forecasting (Yu et al., 2025; Sun et al., 2021). The rapid development of machine learning has driven significant advances in time series forecasting(Yu et al., 2023a; Shao et al., 2024; Sun et al., 2025). Recently, large language models (LLMs) (Radford, 2018) have demonstrated remarkable capabilities in capturing sequential structures and patterns, which is crucial for modeling time-dependent data in time series forecasting. Given the similarities between time series and natural language in

---

[*]Equal contribution   [1]Tianjin University, Tianjin, China [2]Meituan, Beijing, China [3]Xiaohongshu, Beijing, China. Correspondence to: Zongxia Xie <caddiexie@hotmail.com>.

*Proceedings of the 42nd International Conference on Machine Learning*, Vancouver, Canada. PMLR 267, 2025. Copyright 2025 by the author(s).

sequence modeling, several approaches have successfully leveraged pre-trained language models for time series forecasting, yielding promising outcomes (Xue & Salim, 2023; Gruver et al., 2024; Pan et al., 2024; Jia et al., 2024).

However, as shown in Figure 1, arising from the inherent differences between time series and natural language, applying LLMs to time series forecasting presents three primary challenges. Firstly, **cross-domain generalization**. Unlike natural language with domain-consistent structural/semantic rules, time series exhibit diverse statistical patterns across domains. Moreover, identical values may carry domain-specific meanings, hindering effective multi-domain integration into LLMs. To address this challenge, some methods leverage semantic information embedded in datasets as prompts, enabling LLMs to effectively differentiate and adapt to various domains (Liu et al., 2024d). Secondly, **cross-modality alignment**. Pre-trained on an extensive text corpus, LLMs exhibit limited capacity for directly understanding time series, thereby rendering cross-modal alignment a formidable challenge (Zhou et al., 2023). While language-as-prefixes models concatenate the two modalities, the lack of meaningful interaction hinders seamless alignment (Jin et al., 2023).

In addition, **error accumulation in autoregressive frameworks**. For tasks with varying prediction horizons, autoregressive models allow a single framework to handle multiple horizons, avoiding the need for separate training protocols (Liu et al., 2024f;e; Yu et al., 2024). However, supervised training is limited to optimizing a model's ability to predict the next step based on actual historical observations. As a result, it fails to alleviate the adverse effects of error accumulation in autoregressive prediction.

To address these challenges, we propose **LangTime**, a **lan**guage-**g**uided unified model for **time** series forecasting that incorporates cross-domain pre-training and reinforcement learning-based fine-tuning. Specifically, we design Temporal Comprehension Prompts (TCPs) to integrate semantic information and channel details to help LLMs differentiate time series across various domains. Additionally, we condense the time series data into a single token and introduce a reconstruction task to enhance the understanding of the temporal patterns of LLM. Furthermore, we propose TimePPO, a fine-tuning algorithm based on Proximal Policy Optimization (PPO), which mitigates error accumulation during testing and improves long-term forecasting performance. Our contributions are summarized as follows:

- We propose LangTime, an autoregressive model that integrates Temporal Comprehension Prompts to provide domain and channel-specific information, enabling LLMs to better understand and forecast time series.

- We introduce TimePPO, a novel fine-tuning algorithm that alleviates error accumulation in autoregressive pre-

dictions, improving long-term forecasting accuracy.

- Extensive experiments demonstrate that LangTime achieves state-of-the-art performance on widely recognized benchmarks and exhibits strong transferability to unseen domains. Our code are publicly available at: https://github.com/niuwz/LangTime.

## 2. Related Work

### 2.1. Large Language Models for Time Series

LLMs have shown considerable promise through pre-training on data spanning various domains (Doddapaneni et al., 2021; Taylor et al., 2022; Zhan et al., 2024). Nonetheless, foundational models for time series face substantial challenges in aligning multi-domain data due to differences in channel numbers, sampling frequencies, and patterns. Existing works address this issue using tailored strategies. TTM (Ekambaram et al., 2024) employs frequency prefixes, Lag-Llama (Rasul et al., 2023) uses lag features as covariates for probabilistic univariate forecasting, UniTime (Liu et al., 2024d) introduces masking and domain instructions to mitigate convergence imbalances, and ROSE (Wang et al., 2024) adopts frequency-based masking and reconstruction to unify cross-domain representations.

While multi-domain approaches focus on aligning time series data across sources, recent methods explore integrating textual information to further enhance forecasting capabilities, introducing a new challenge of cross-modality alignment. Directly concatenating time series and language tokens often lead to a modality gap due to structural and semantic differences. To address this, CALF (Liu et al., 2024c) applies knowledge distillation, $S^2$IP-LLM (Pan et al., 2024) aligns semantic and time series spaces with tokenization and anchors, and TimeCMA (Liu et al., 2024a) uses prompt-based techniques to extract time series representations.

However, existing methods address these challenges in isolation, leaving a gap in simultaneously solving multi-domain and cross-modality alignment. LangTime bridges this gap by tackling both challenges concurrently. By facilitating robust cross-modality alignment, LangTime achieves superior performance in time series forecasting across diverse domains.

### 2.2. Reinforcement Learning from Human Feedback in Large Language Models

Reinforcement Learning from Human Feedback (RLHF) has gained widespread application and demonstrated remarkable success in Natural Language Processing (NLP) tasks. For example, InstructGPT (Ouyang et al., 2022) uses human preference data to train a Reward Model (RM), which is subsequently employed to fine-tune the supervised policy using the Proximal Policy Optimization (PPO) algo-

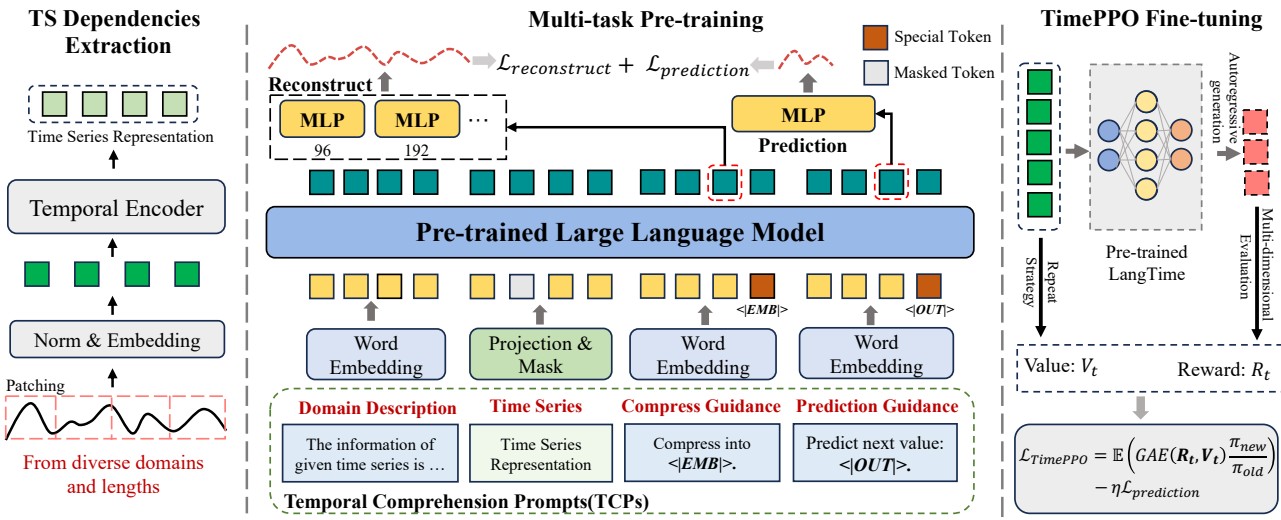

Figure 2. Overview of LangTime: (1) The Temporal Encoder extracts temporal dependencies from time series data of varying lengths and domains, generating unified time series representations. (2) Temporal Comprehension Prompts are constructed by integrating time series representations with domain-specific context, enabling LLMs to perform temporal pattern compression. This pre-training phase establishes modality alignment by jointly optimizing reconstruction and prediction tasks. (3) During fine-tuning, the TimePPO algorithm is implemented to mitigate error accumulation in autoregressive forecasting, thereby enhancing long-term prediction. The content of the prompt has been simplified in the figure, with the detailed information provided in Section 3.3 and Appendix A.2.

rithm (Schulman et al., 2017). This framework effectively enhances model performance by aligning outputs with human preferences. However, the requirement to train and optimize a reward model introduces significant computational overhead, making PPO resource-intensive in practical scenarios.

Despite the success of RLHF in NLP, its application to time series tasks remains underexplored, especially for autoregressive models. Time series forecasting introduces unique error accumulation challenges that require significant adaptation of existing RLHF methods. To bridge this gap, we adapt the PPO algorithm to the specific demands of time series prediction tasks. We redesign the reward function and value function to better capture the temporal structure and mitigate error propagation, simplifying the algorithm while enhancing its applicability to complex temporal data. These innovations make PPO more efficient and effective for autoregressive time series forecasting, addressing challenges such as computational complexity and scalability in long-term predictions.

## 3. Methodology

**Problem Definition.** Given the historical observations of multivariate time series $\mathbf{X}_t = \{\mathbf{x}^i_{t-L:t}\}^C_{i=1}$, where $L$ represents the number of lookback time steps and $C$ denotes the number of variates, the goal is to predict the future $F$ time steps $\hat{\mathbf{Y}}_t = \{\hat{\mathbf{x}}^i_{t:t+F}\}^C_{i=1}$. The ground truth of the future values is denoted as $\mathbf{Y}_t = \{\mathbf{x}^i_{t:t+F}\}^C_{i=1}$. LangTime is pre-trained on multi-source datasets $\mathcal{D}_{\text{pre-train}} = \{(\mathbf{X}^j_t, \mathbf{Y}^j_t)\}^N_{j=1}$,

where $N$ is the number of datasets. For downstream task in dataset $j$, the model is fine-tuned on $\mathcal{D}^j_{\text{fine-tune}}$ and tested on $\mathcal{D}^j_{\text{test}}$. The datasets for pre-training, fine-tuning, and testing are pairwise disjoint to ensure generalization.

### 3.1. Architecture

LangTime includes three novel components: the Temporal Encoder (TE), Temporal Comprehension Prompts (TCPs), and Time Series Proximal Policy Optimization (TimePPO), as illustrated in Figure 2. To extract meaningful representations, LangTime employs the TE to process continuous time series. To bridge the gap between time series and language, TCPs encode essential contextual information, enabling LLMs to effectively differentiate and interpret time series data. The processed time series representations and contextual information are then passed into a pre-trained LLM. To further improve forecasting stability, LangTime incorporates TimePPO, a reinforcement learning-based fine-tuning strategy designed to mitigate error accumulation and enhance multi-step prediction. These components work together to ensure effective alignment between time series and language and enhance long-term forecasting performance.

**Multi-task Pre-training.** To align time series representation with the word embedding space, we utilize two pre-training tasks: reconstruction and prediction. The reconstruction task enhances the model's understanding of time series, while the prediction task leverages the generative capabilities of LLMs to identify anticipatory dependencies for future forecasting (Cao et al., 2020). We adopt the Hu-

ber loss (Huber, 1992) to balance robustness and sensitivity. The overall pre-training loss is formulated as:

$$\mathcal{L}_{\text{pre-train}} = \alpha\mathcal{L}_{\text{reconstruction}} + (1 - \alpha)\mathcal{L}_{\text{prediction}}, \quad (1)$$

where $\alpha$ controls the trade-off between the reconstruction and prediction tasks. Details about the loss function can be found in Appendix A.1.

**Input Processing and Adaptive Training.** The input sequence $X \in \mathbb{R}^{L \times C}$ is first divided into non-overlapping patches of length $P$. For each patch $\mathcal{P}_i \in \mathbb{R}^{C \times P}$, we apply a linear transformation to project it into $\mathcal{P}'_i \in \mathbb{R}^{C \times D}$.

To enable long-term forecasting in an autoregressive manner, our model maintains a fixed output sequence length while allowing the input sequence length to vary, constrained to integer multiples of the patch size. This design exposes the model to diverse input lengths, improving its generalization ability for autoregressive predictions. To address the imbalance caused by longer input sequences overshadowing shorter ones, we propose a progressive training strategy. This approach transitions from short to long sequence data, allowing the model to gradually adapt to the complexities of longer sequences.

### 3.2. Temporal Encoder

To align time series with LLMs, we introduce a channel-independent lightweight TE (Nie et al., 2022) to capture temporal dependencies and generate rich time series representations, which are mapped into the word embedding space using a simple linear layer (Liu et al., 2024b):

$$F = \text{Linear}(\text{TE}(\mathcal{P}')). \quad (2)$$

Time series datasets exhibit varying convergence rates due to differences in their underlying statistical properties (Liu et al., 2024d). To address this, we introduce a random masking strategy to enhance the generalization across diverse datasets of LLMs. Specifically, we use a learnable token $\mathcal{P}_m$ to randomly replace patches in $F$ at a proportion of $r_m$, thereby obtaining $F_m$. During training, portions of the time series representation are randomly masked, prompting the model to infer missing values and capture deeper temporal structures for the reconstruction task. This strategy not only strengthens the model's robustness but also improves its adaptability to datasets with distinct characteristics.

### 3.3. Temporal Comprehension Prompts

To fully leverage the generative capabilities of LLMs, we introduce TCPs specifically designed to bridge the gap between time series and language models. TCPs align time series representations with the structural characteristics of LLMs, facilitating both comprehension and forecasting. A detailed illustration of TCPs is provided in Appendix A.2.

---

**Temporal Comprehension Prompts**

The information of the given time series:
Period: <Timestamp>,
Dataset: <Dataset Information>,
Channel: <Channel Information>,
Value: <Time Series Representation>,
Please compress this series into one word: <|EMB|>.
Based on the given information, predict next <N> values:
<|OUT|>.

---

TCPs play a crucial role in guiding the LLM's interpretation of time series data by encoding essential contextual information. The domain description encodes dataset-specific characteristics, allowing the model to incorporate relevant contextual information. The time placeholder is replaced with time series features extracted by TE to provide direct temporal input. Additionally, compression token enables the condensation of time series information into a single token, enhancing the model's ability to capture key temporal patterns. The prediction guidance directs the forecasting process, ensuring smooth integration between time series data and the LLM's generative structure.

All components of the TCPs are fed into the pre-trained LLM, where the causal attention mechanism enables each token to incorporate information from all preceding tokens. Following (Zhang et al., 2024), we extract the previous token from <|EMB|> and <|OUT|> respectively, serving as the compressed token and prediction token. Ultimately, we employed a projection operation to obtain the reconstruction and prediction:

$$\hat{X} = \text{Linear}(\text{LLM}(F_m)[\text{index}(<|\text{EMB}|>) - 1]), \quad (3)$$

$$\hat{Y} = \text{Linear}(\text{LLM}(F_m)[\text{index}(<|\text{OUT}|>) - 1]). \quad (4)$$

The compressed token serves as a global summary of the time series and is used for reconstruction, while distinct linear layers handle variable sequence lengths, allowing the model to generalize across different history lengths. Under the guidance of TCPs, LLMs forecast future values by referencing both temporal embeddings and the compressed token, ensuring seamless integration of temporal and linguistic representations. This design enhances alignment between time series and LLMs, allowing LangTime to leverage LLMs' strengths in sequential modeling while enhancing forecasting performance.

### 3.4. Time Series Proximal Policy Optimization

Autoregressive methods often suffer from high variance due to error propagation, which significantly degrades performance in long-term forecasting compared to non-autoregressive methods (Taieb & Atiya, 2015). To address this issue, we propose TimePPO, an extension of the PPO algorithm (Schulman et al., 2017) tailored for time series

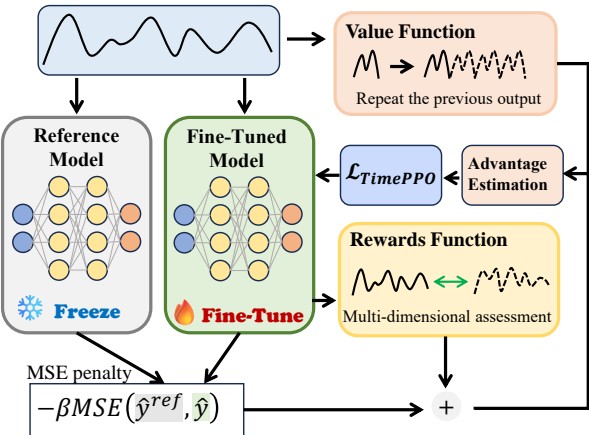

*Figure 3.* Overview of TimePPO. Both the *Reference Model* and the *Fine-Tuned Model* are initialized with the same parameters. The *Value Function* uses a repeat strategy to estimate expected returns, while the *Rewards Function* evaluates the accuracy of prediction results through multiple dimensions.

forecasting. Considering the unique characteristics of time series, we have undertaken a redesign of the Rewards Function and Value Function.

**Rewards Function.** As shown in Figure 3, the rewards function evaluates the discrepancy between the predicted series $\hat{y}_t$ and the ground truth $y_t$ at time $t$ from multiple perspectives, Additionally, we introduce an MSE penalty term $MSE(\hat{y}_t, \hat{y}_t^{\text{ref}})$ to constrain the difference between the new policy and the old policy, ensuring stability and safety during policy updates, as defined in Equation (5):

$$R(\hat{y}_t) = \tanh\left(\tau \sum_i \mathcal{R}_i(\hat{y}_t, y_t)w_i\right) - \beta MSE(\hat{y}_t, \hat{y}_t^{\text{ref}}),$$
(5)

where $\tau$ is a hyperparameter controlling the distribution of the reward scores, and $\mathcal{R}_i(\hat{y}_t, y_t)$ represents individual evaluation dimension (detailed in Appendix A.3). The weights $w_i$ balance the contributions of these dimensions.

$$V(x_t) = \begin{cases} 0, & \text{if } t = 0 \\ \sum_{i=t}^{m}\left(\tanh\left(\tau \sum_j \mathcal{R}_i(\hat{y}_{t-1}, y_i)w_j\right)\right). & \text{if } t > 0 \end{cases}$$
(6)

**Value Function.** We define the value function $V(x_t)$ to estimate the expected return from state $x_t$. As shown in Equation (6), we assume that the model will continue to repeat the output from time $t - 1$ until the final time $m$, and then consider all the reward scores obtained as the expected return for predicting the entire sequence. This design ensures that the value function reflects the accumulated rewards, facilitating more accurate return estimation over time.

**Advantage Estimation.** To estimate the advantage, we employ Generalized Advantage Estimation (GAE) (Schulman et al., 2015):

$$\hat{A}_t = \delta_t + (\gamma\lambda)\delta_{t+1} + \cdots + (\gamma\lambda)^{T-t+1}\delta_{T-1},$$
$$\delta_t = r_t + \gamma V(x_{t+1}) - \xi V(x_t),$$
(7)

where $\xi$ is introduced to control the sensitivity to estimation errors in the current state's value, thereby mitigating the error accumulation.

**TimePPO.** Inspired by InstructGPT, we introduce an alignment tax to prevent performance degradation (Ouyang et al., 2022). The loss function for TimePPO is given as:

$$\mathcal{L}_{\text{TimePPO}} = \mathbb{E}\Big[\min\Big(r(\theta)\hat{A}(x, y),$$
$$\text{clip}(r(\theta), 1 - \epsilon, 1 + \epsilon)\hat{A}(x, y)\Big)\Big]$$
$$- \eta\mathcal{L}_{\text{prediction}},$$
(8)

where $r(\theta)$ represents the policy ratio, and $\epsilon$ regulates the extent of policy updates to ensure stability. The alignment tax term $\eta\mathcal{L}_{\text{prediction}}$ penalizes large deviations from the ground truth, ensuring that the model remains aligned with accurate predictions. Further details are presented in Appendix A.4.

With the aforementioned design, we have successfully enhanced the PPO algorithm for time series, thereby optimizing the autoregressive prediction model from an entirely novel perspective. The overall process of the TimePPO is depicted in Algorithm 1.

---
**Algorithm 1** TimePPO

---
1: Input: initial policy parameters $\theta_0$ and $\theta_{\text{ref}}$.
2: **for** $k = 0, 1, 2, ...$ **do**
3:     Collect set of trajectories $\mathcal{D}_k = \{x_t, \hat{y}_t, \hat{y}_t^{\text{ref}}\}$ by running policy $\pi_k = \pi(\theta_k)$ and $\pi_{\text{ref}} = \pi(\theta_{\text{ref}})$.
4:     Compute rewards $R(\hat{y}_t)$ and values $V(x_{t-1})$.
5:     Compute advantage estimates $\hat{A}_t$.
6:     Update the policy $\pi_k = \pi(\theta_k)$ by maximizing the TimePPO objective in Equation (8).
7: **end for**

---

## 4. Experiments

### 4.1. Training Details

**Datasets.** We conduct both pre-training and fine-tuning of LangTime on seven real-world datasets that cover various time series application domains, including ETT (ETTh1, ETTh2, ETTm1, ETTm2), Electricity, Exchange, and Weather. Specifically, we pre-train LangTime across multiple domains and then perform fine-tuning on individual domain using the TimePPO algorithm. We evaluate the performance of LangTime on the test splits of the same

*Table 1.* Forecasting performance comparisons. The input sequence length is set to 96. The predictive lengths are set to 96, 192, 336, 720. Avg is averaged over all predictive lengths. **Red**: best performance for the entire row. **Bold**: best performance among models trained across datasets.

| Method | | Models Trained Across Datasets | | | | | | Models Trained/Fine-tuned on Each Dataset | | | | | | | | | | | | | | |
|---|---|---|---|---|---|---|---|---|---|---|---|---|---|---|---|---|---|---|---|---|---|---|---|
| | | LangTime$_{PT}$ | | UniTime† | | AutoTimes† | | LangTime$_{TimePPO}$ | | UniTime$^‡_{SFT}$ | | AutoTimes* | | S²IP-LLM* | | Time-LLM* | | GPT4TS* | | PatchTST | | TimesNet | |
| | | MSE | MAE | MSE | MAE | MSE | MAE | MSE | MAE | MSE | MAE | MSE | MAE | MSE | MAE | MSE | MAE | MSE | MAE | MSE | MAE | MSE | MAE |
| ETTm1 | 96 | **0.323** | **0.346** | 0.350 | 0.385 | 0.914 | 0.590 | **0.319** | 0.348 | 0.337 | 0.374 | 0.530 | 0.519 | 0.359 | 0.381 | 0.327 | 0.361 | 0.327 | 0.363 | 0.329 | 0.367 | 0.338 | 0.375 |
| | 192 | **0.372** | **0.376** | 0.384 | 0.401 | 0.966 | 0.616 | 0.368 | **0.375** | 0.373 | 0.393 | 0.664 | 0.583 | 0.383 | 0.393 | 0.368 | 0.381 | 0.368 | 0.383 | **0.367** | 0.385 | 0.374 | 0.387 |
| | 336 | **0.419** | **0.403** | 0.420 | 0.423 | 0.935 | 0.612 | 0.413 | **0.402** | 0.405 | 0.415 | 0.836 | 0.652 | 0.416 | 0.414 | 0.401 | 0.405 | 0.400 | 0.405 | **0.399** | 0.410 | 0.410 | 0.411 |
| | 720 | 0.491 | **0.443** | **0.473** | 0.455 | 0.954 | 0.630 | 0.487 | 0.439 | 0.465 | 0.448 | 1.297 | 0.782 | 0.483 | 0.449 | 0.465 | **0.436** | 0.461 | 0.439 | **0.454** | 0.439 | 0.478 | 0.450 |
| | Avg | **0.401** | **0.392** | 0.407 | 0.416 | 0.942 | 0.612 | 0.397 | **0.391** | 0.395 | 0.408 | 0.832 | 0.634 | 0.410 | 0.409 | 0.390 | 0.396 | 0.389 | 0.398 | **0.387** | 0.400 | 0.400 | 0.406 |
| ETTm2 | 96 | 0.184 | **0.258** | **0.183** | 0.264 | 0.269 | 0.331 | 0.188 | **0.258** | 0.179 | 0.264 | 0.722 | 0.588 | 0.193 | 0.280 | 0.177 | 0.262 | 0.178 | 0.264 | **0.175** | 0.259 | 0.187 | 0.267 |
| | 192 | **0.245** | **0.300** | 0.246 | 0.307 | 0.326 | 0.364 | 0.245 | **0.297** | 0.246 | 0.307 | 0.898 | 0.673 | 0.257 | 0.318 | 0.245 | 0.305 | 0.246 | 0.307 | **0.241** | 0.302 | 0.249 | 0.309 |
| | 336 | **0.308** | **0.339** | 0.310 | 0.346 | 0.379 | 0.393 | 0.301 | **0.336** | 0.308 | 0.346 | 1.207 | 0.792 | 0.317 | 0.353 | 0.304 | 0.342 | 0.309 | 0.349 | 0.305 | 0.343 | 0.321 | 0.351 |
| | 720 | **0.410** | **0.399** | 0.413 | 0.405 | 0.473 | 0.442 | 0.402 | **0.393** | 0.410 | 0.406 | 1.708 | 0.952 | 0.419 | 0.411 | **0.400** | 0.397 | 0.410 | 0.408 | 0.402 | 0.400 | 0.408 | 0.403 |
| | Avg | **0.287** | **0.324** | 0.288 | 0.331 | 0.362 | 0.383 | 0.284 | **0.321** | 0.286 | 0.331 | 1.134 | 0.751 | 0.297 | 0.341 | 0.282 | 0.327 | 0.286 | 0.332 | **0.281** | 0.326 | 0.291 | 0.333 |
| ETTh1 | 96 | **0.394** | **0.395** | 0.525 | 0.500 | 0.417 | 0.408 | 0.391 | **0.388** | 0.480 | 0.473 | **0.381** | 0.401 | 0.398 | 0.410 | 0.423 | 0.425 | 0.385 | 0.402 | 0.414 | 0.419 | 0.384 | 0.402 |
| | 192 | **0.439** | **0.420** | 0.544 | 0.511 | 0.484 | 0.444 | **0.429** | **0.419** | 0.505 | 0.487 | 0.435 | 0.434 | 0.451 | 0.440 | 0.464 | 0.446 | 0.432 | 0.425 | 0.460 | 0.445 | 0.436 | 0.429 |
| | 336 | **0.464** | **0.442** | 0.576 | 0.529 | 0.529 | 0.468 | **0.462** | **0.440** | 0.536 | 0.504 | 0.480 | 0.459 | 0.508 | 0.471 | 0.499 | 0.461 | 0.467 | 0.447 | 0.501 | 0.466 | 0.491 | 0.469 |
| | 720 | **0.462** | **0.449** | 0.577 | 0.548 | 0.549 | 0.494 | **0.458** | **0.445** | 0.531 | 0.520 | 0.499 | 0.478 | 0.483 | 0.478 | 0.505 | 0.487 | 0.472 | 0.466 | 0.500 | 0.488 | 0.521 | 0.500 |
| | Avg | **0.440** | **0.427** | 0.556 | 0.522 | 0.495 | 0.454 | **0.435** | **0.423** | 0.513 | 0.496 | 0.449 | 0.443 | 0.460 | 0.450 | 0.473 | 0.455 | 0.439 | 0.435 | 0.469 | 0.455 | 0.458 | 0.450 |
| ETTh2 | 96 | 0.301 | 0.334 | 0.308 | 0.357 | **0.301** | **0.333** | 0.299 | 0.336 | 0.306 | 0.356 | 0.318 | 0.352 | **0.295** | 0.346 | 0.306 | 0.354 | 0.303 | 0.354 | 0.302 | 0.348 | 0.340 | 0.374 |
| | 192 | **0.380** | **0.389** | 0.390 | 0.405 | 0.410 | 0.398 | **0.374** | **0.382** | 0.388 | 0.404 | 0.401 | 0.404 | 0.386 | 0.399 | 0.377 | 0.398 | 0.386 | 0.404 | 0.388 | 0.400 | 0.402 | 0.414 |
| | 336 | **0.412** | **0.419** | 0.424 | 0.438 | 0.420 | **0.419** | **0.410** | **0.418** | 0.423 | 0.436 | 0.446 | 0.441 | 0.447 | 0.443 | 0.423 | 0.435 | 0.430 | 0.438 | 0.426 | 0.433 | 0.452 | 0.452 |
| | 720 | **0.422** | **0.439** | 0.435 | 0.454 | 0.439 | 0.444 | **0.418** | **0.426** | 0.433 | 0.453 | 0.460 | 0.459 | 0.428 | 0.444 | 0.431 | 0.447 | 0.433 | 0.452 | 0.431 | 0.446 | 0.462 | 0.468 |
| | Avg | **0.379** | **0.395** | 0.389 | 0.414 | 0.393 | 0.399 | **0.375** | **0.391** | 0.388 | 0.412 | 0.406 | 0.414 | 0.389 | 0.408 | 0.384 | 0.409 | 0.388 | 0.412 | 0.387 | 0.407 | 0.414 | 0.427 |
| Electricity | 96 | 0.199 | 0.277 | 0.279 | 0.382 | **0.188** | **0.267** | 0.181 | 0.266 | 0.210 | 0.381 | 0.206 | 0.277 | 0.204 | 0.293 | 0.184 | 0.268 | 0.184 | 0.270 | 0.181 | 0.270 | **0.168** | 0.272 |
| | 192 | **0.213** | 0.296 | 0.276 | 0.379 | 0.217 | **0.291** | 0.185 | 0.273 | 0.249 | 0.351 | 0.224 | 0.296 | 0.207 | 0.295 | 0.204 | 0.286 | 0.188 | 0.275 | 0.188 | 0.274 | **0.184** | 0.289 |
| | 336 | **0.234** | **0.316** | 0.285 | 0.385 | 0.236 | 0.319 | 0.198 | 0.281 | 0.259 | 0.357 | 0.251 | 0.322 | 0.219 | 0.308 | 0.222 | 0.308 | 0.203 | 0.290 | 0.204 | 0.293 | **0.198** | 0.300 |
| | 720 | **0.272** | 0.357 | 0.322 | 0.409 | **0.272** | 0.346 | 0.241 | 0.320 | 0.298 | 0.362 | 0.318 | 0.380 | 0.263 | 0.341 | 0.269 | 0.345 | 0.243 | 0.322 | 0.246 | 0.324 | **0.220** | **0.320** |
| | Avg | 0.230 | 0.312 | 0.291 | 0.389 | **0.228** | **0.306** | 0.201 | 0.285 | 0.254 | 0.363 | 0.250 | 0.319 | 0.223 | 0.309 | 0.220 | 0.302 | 0.205 | 0.289 | 0.205 | 0.290 | **0.193** | 0.295 |
| Exchange | 96 | **0.086** | **0.205** | 0.124 | 0.254 | 0.133 | 0.253 | 0.089 | **0.201** | 0.118 | 0.246 | 0.087 | 0.202 | **0.083** | 0.203 | 0.087 | 0.208 | 0.084 | **0.201** | 0.088 | 0.205 | 0.107 | 0.234 |
| | 192 | **0.175** | **0.300** | 0.218 | 0.338 | 0.253 | 0.357 | **0.175** | **0.298** | 0.212 | 0.332 | 0.178 | **0.298** | 0.178 | 0.299 | 0.178 | 0.302 | 0.178 | 0.299 | 0.176 | 0.299 | 0.226 | 0.344 |
| | 336 | **0.329** | **0.412** | 0.367 | 0.443 | 0.390 | 0.452 | 0.329 | 0.409 | 0.360 | 0.438 | 0.328 | 0.413 | 0.338 | 0.415 | 0.338 | 0.422 | 0.343 | 0.422 | **0.301** | **0.397** | 0.367 | 0.448 |
| | 720 | **0.854** | **0.696** | 0.913 | 0.728 | 0.931 | 0.730 | 0.852 | 0.690 | 0.904 | 0.723 | **0.792** | 0.675 | 0.854 | 0.696 | 0.819 | 0.681 | 0.803 | **0.671** | 0.901 | 0.714 | 0.964 | 0.746 |
| | Avg | **0.361** | **0.403** | 0.406 | 0.441 | 0.427 | 0.448 | 0.361 | 0.400 | 0.399 | 0.435 | **0.347** | **0.397** | 0.361 | 0.403 | 0.356 | 0.403 | 0.352 | 0.398 | 0.367 | 0.404 | 0.416 | 0.443 |
| Weather | 96 | 0.184 | **0.203** | **0.181** | 0.222 | 0.219 | 0.245 | 0.178 | **0.202** | 0.182 | 0.220 | 0.188 | 0.227 | 0.195 | 0.233 | 0.180 | 0.221 | 0.183 | 0.223 | 0.177 | 0.218 | **0.172** | 0.220 |
| | 192 | **0.216** | **0.250** | 0.226 | 0.261 | 0.298 | 0.310 | **0.211** | **0.245** | 0.226 | 0.263 | 0.234 | 0.266 | 0.240 | 0.269 | 0.229 | 0.261 | 0.230 | 0.262 | 0.218 | 0.259 | 0.219 | 0.261 |
| | 336 | **0.275** | **0.293** | 0.280 | 0.299 | 0.337 | 0.338 | **0.269** | **0.286** | 0.279 | 0.323 | 0.288 | 0.305 | 0.293 | 0.306 | 0.285 | 0.301 | 0.285 | 0.302 | 0.278 | 0.297 | 0.280 | 0.306 |
| | 720 | 0.361 | 0.349 | **0.356** | **0.347** | 0.415 | 0.383 | **0.351** | 0.346 | 0.354 | **0.346** | 0.363 | 0.355 | 0.368 | 0.354 | 0.359 | 0.349 | 0.362 | 0.351 | 0.354 | 0.348 | 0.365 | 0.359 |
| | Avg | **0.259** | **0.274** | 0.261 | 0.282 | 0.317 | 0.319 | **0.252** | **0.270** | 0.260 | 0.288 | 0.268 | 0.288 | 0.274 | 0.291 | 0.263 | 0.283 | 0.265 | 0.285 | 0.257 | 0.281 | 0.259 | 0.287 |
| 1st Count | | 58 | | 5 | | 10 | | 45 | | 1 | | 5 | | 2 | | 2 | | 2 | | 9 | | 7 | |

† signifies the use of the official baseline code with cross-domain training conducted similarly to our approach.

‡ denotes supervised fine-tuning using identical data as our method, building upon the conditions specified by †.

∗ indicates the adoption of the official baseline code, with adjustments to input sequence length and maximum training epochs for fair comparison with other methods, and other results are sourced from iTransformer(Liu et al., 2023).

seven benchmark datasets. The details of these datasets are provided in Appendix B.1.

**Baselines.** We evaluate LangTime against state-of-the-art models. (1) **LLM-based methods**: UniTime (Liu et al., 2024d), AutoTimes (Liu et al., 2024e), S²IP-LLM (Pan et al., 2024), Time-LLM (Jin et al., 2023), and GPT4TS (Zhou et al., 2023); (2) **Specific methods**: PatchTST (Nie et al., 2022), and TimesNet (Wu et al., 2022). We adopt Qwen2-0.5B-Instruction (Yang et al., 2024) as backbone.

**Setup.** We adopt pre-training to fine-tuning dataset ratio of $\mathcal{D}_{\text{pre-train}} : \mathcal{D}_{\text{fine-tune}} = 8 : 2$. To ensure a fair comparison, all methods maintained a fixed look-back window of $L = 96$ and predicted future values with lengths of $F = \{96, 192, 336, 720\}$. More implementation details can be found in Appendix B.2.

### 4.2. Main Results

**Comparison with Other Forecasting Methods.** Table 1 presents the overall forecasting performance of our model.

The table is divided into two sections by vertical lines. On the left, models are pre-trained across multiple datasets, while on the right, models are either trained separately or fine-tuned for each dataset.

Pre-trained LangTime achieves remarkable improvements over baseline models that are also trained across datasets, achieving the best performance in 58 out of 70 entries. This highlights LangTime's ability to generalize effectively across diverse time series distributions. On the right side of the table, results indicate that TimePPO fine-tuned Lang-Time achieves competitive performance, surpassing models trained individually on each dataset in 45 out of 70 entries, establishing new state-of-the-art results. These findings validate LangTime's effectiveness in handling time series data with diverse characteristics. Furthermore, compared to other LLM-based forecasting methods, LangTime demonstrates significant advantages, reinforcing its capability to enhance the alignment between LLMs and time series data.

**Comparison with Other Fine-tuning Algorithms.** Table 2 presents a comparison between our proposed TimePPO algo-

*Table 2.* Comparisons of forecasting performance among various fine-tuning algorithms. The results are presented as averages over four forecasting horizons: 96, 192, 336, and 720. **Bold**: best results. Table 18 shows the full results.

| Method | ETTm1 | | ETTm2 | | ETTh1 | | ETTh2 | | Electricity | | Exchange | | Weather | |
|---|---|---|---|---|---|---|---|---|---|---|---|---|---|---|
| | MSE | MAE | MSE | MAE | MSE | MAE | MSE | MAE | MSE | MAE | MSE | MAE | MSE | MAE |
| LangTime | 0.401 | 0.392 | 0.287 | 0.324 | 0.440 | 0.427 | 0.379 | 0.395 | 0.230 | 0.312 | 0.361 | 0.403 | 0.259 | 0.274 |
| LangTime$_{SFT}$ | 0.399 | 0.391 | 0.285 | 0.321 | 0.447 | 0.423 | 0.378 | 0.391 | 0.211 | 0.291 | 0.362 | 0.404 | 0.263 | 0.279 |
| LangTime$_{TimePPO}$ | **0.397** | 0.391 | **0.284** | 0.321 | **0.435** | **0.423** | **0.375** | 0.391 | **0.201** | **0.285** | 0.361 | **0.400** | **0.252** | **0.270** |
| AutoTimes | 0.942 | 0.612 | 0.362 | 0.383 | 0.495 | 0.454 | 0.392 | 0.399 | 0.228 | 0.306 | 0.427 | 0.448 | 0.317 | 0.319 |
| AutoTimes$_{SFT}$ | 0.945 | 0.613 | 0.364 | 0.383 | 0.491 | 0.448 | 0.400 | 0.403 | **0.228** | 0.307 | **0.424** | **0.444** | 0.318 | 0.319 |
| AutoTimes$_{TimePPO}$ | **0.940** | **0.610** | **0.360** | 0.383 | **0.485** | **0.446** | **0.390** | 0.399 | 0.233 | **0.304** | 0.425 | 0.450 | **0.317** | **0.318** |

*Table 3.* Ablation studies on various components of temporal comprehension prompts on ETTh1 and Weather datasets. Results are reported as averages over four forecasting horizons: 96, 192, 336, and 720. Full results are accessible in Table 19.

| LG | TS | DI | CI | ETTh1 | | Weather | |
|---|---|---|---|---|---|---|---|
| | | | | MSE | MAE | MSE | MAE |
| ✓ | ✓ | ✓ | ✓ | **0.436** | **0.436** | **0.267** | **0.284** |
| ✓ | ✓ | ✓ | | 0.440 | 0.438 | 0.269 | 0.288 |
| ✓ | ✓ | | | 0.442 | 0.440 | 0.272 | 0.292 |
| ✓ | | | | 0.442 | 0.443 | 0.274 | 0.293 |

*Table 4.* Ablation studies on various dimensions of Rewards Function on ETTh1 and Weather datasets. Results are reported as averages over four forecasting horizons: 96, 192, 336, and 720. Full results are accessible in Table 20.

| Method | ETTh1 | | Weather | |
|---|---|---|---|---|
| | MSE | MAE | MSE | MAE |
| LangTime$_{PT}$ | 0.436 | 0.436 | 0.267 | 0.284 |
| All dimensions | **0.429** | **0.433** | **0.259** | **0.280** |
| TimePPO w/o $\mathcal{R}_{MSE}$ | 0.435 | 0.437 | 0.263 | 0.281 |
| TimePPO w/o $\mathcal{R}_{MAE}$ | 0.433 | 0.438 | 0.261 | 0.282 |
| TimePPO w/o $\mathcal{R}_{KL}$ | 0.435 | 0.438 | 0.262 | 0.282 |

rithm and Supervised Fine-tuning algorithm (SFT). Across most datasets, models fine-tuned with TimePPO achieve superior predictive performance compared to those fine-tuned with SFT. This demonstrates the effectiveness of TimePPO in mitigating cumulative errors and enhancing the stability of autoregressive forecasting models.

### 4.3. Ablation Studies

**Temporal Comprehension Prompts.** We analyzed the impact of TCPs at different levels of detail by segmenting TCPs into the following components: ① **Language Guidance (LG)**; ② **Timestamp (TS)**; ③ **Dataset Information (DI)**; ④ **Channel Information (CI)**. The detailed results in Table 3 show that removing CI and DI leads to a decline in LangTime's performance, while the impact of TS is relatively minor. Notably, even in the absence of supplementary metadata, using only LG still yields favorable results, further confirming the effectiveness of our proposed approach in aligning LLMs with time series data.

**Rewards Function.** We comprehensively evaluate the ac-

*Table 5.* Ablation study of different backbones on ETTh1 and Weather datasets. **Bold**: best results.

| Dataset | | GPT2 | | Qwen | | Linear | |
|---|---|---|---|---|---|---|---|
| | | MSE | MAE | MSE | MAE | MSE | MAE |
| ETTh1 | 96 | 0.409 | 0.406 | **0.388** | **0.391** | 0.702 | 0.567 |
| | 192 | 0.472 | 0.438 | **0.442** | **0.423** | 0.721 | 0.580 |
| | 336 | 0.530 | 0.461 | **0.479** | **0.445** | 0.733 | 0.592 |
| | 720 | 0.556 | 0.483 | **0.482** | **0.465** | 0.735 | 0.611 |
| | Avg | 0.492 | 0.447 | **0.448** | **0.431** | 0.723 | 0.588 |
| Weather | 96 | 0.194 | 0.233 | **0.181** | **0.217** | 0.223 | 0.273 |
| | 192 | 0.244 | 0.276 | **0.236** | **0.254** | 0.263 | 0.310 |
| | 336 | **0.294** | 0.311 | **0.294** | **0.295** | 0.318 | 0.343 |
| | 720 | **0.366** | 0.358 | 0.368 | **0.341** | 0.398 | 0.405 |
| | Avg | 0.275 | 0.294 | **0.270** | **0.277** | 0.301 | 0.333 |

curacy of prediction results from three dimensions in the TimePPO, including Mean Squared Error (MSE), Mean Absolute Error (MAE), and KL divergence, as detailed in Table 10. We evaluate LangTime's predictions across the three different dimensions and compute the corresponding reward scores, with the results presented in Table 4. The findings indicate that incorporating all dimensions significantly enhances the performance of pre-trained LangTime. Conversely, removing any single dimension leads to a decline in TimePPO's effectiveness, with MSE exerting the most significant impact compared to others.

**Backbones.** We selected Qwen2-0.5B-Instruction (Yang et al., 2024) as the backbone due to its superior instruction-following capability. To validate the framework's scalability, we substituted the backbone with GPT-2 (Radford et al., 2019). Table 5 demonstrates that while GPT-2 achieves comparable performance on the Weather dataset, it underperforms on ETTh1 due to weaker instruction compliance. To evaluate LLMs' role in transferring sequential modeling from text to time series, we replaced the backbone with a linear layer (Zeng et al., 2023). As presented in Table 5, this modification resulted in significant performance degradation, conclusively demonstrating that our method effectively enhances LLMs' comprehension of time series patterns.

**Adaptability to Different Prompts.** The TCPs in our approach comprise two components. The background part incorporates domain and channel descriptions to provide richer linguistic information, enabling LLMs to develop a

*Table 6.* Impact of language prompt modification on model capability on ETTh1 and Weather datasets. **Bold**: best performance.

| Dataset | | Original | | Instruction | | Background | |
|---|---|---|---|---|---|---|---|
| | | MSE | MAE | MSE | MAE | MSE | MAE |
| ETTh1 | 96 | 0.388 | **0.391** | **0.386** | 0.397 | 0.387 | 0.396 |
| | 192 | 0.442 | **0.423** | 0.442 | 0.425 | **0.441** | 0.426 |
| | 336 | 0.479 | **0.445** | 0.479 | **0.445** | **0.478** | **0.445** |
| | 720 | **0.482** | 0.465 | 0.488 | **0.465** | 0.493 | 0.468 |
| | Avg | **0.448** | **0.431** | 0.449 | 0.433 | 0.450 | 0.434 |
| Weather | 96 | **0.181** | 0.217 | **0.181** | 0.219 | 0.184 | **0.214** |
| | 192 | **0.236** | **0.254** | 0.241 | 0.261 | 0.237 | 0.256 |
| | 336 | **0.294** | **0.295** | 0.297 | 0.300 | 0.296 | 0.298 |
| | 720 | **0.368** | **0.341** | 0.371 | 0.347 | 0.372 | 0.348 |
| | Avg | **0.270** | **0.277** | 0.273 | 0.282 | 0.272 | 0.279 |

*Table 7.* Comparison of zero-shot performance of pre-trained models. The input sequence length is set to 96 for the Traffic dataset and 48 for the Illness to fit patch size 24. The predictive lengths are set to 24, 36, 48, 60 for Illness, and 96, 192, 336, 720 for Traffic. **Bold**: best results.

| Dataset | | LangTime$_{PT}$ | | UniTime | | AutoTimes | |
|---|---|---|---|---|---|---|---|
| | | MSE | MAE | MSE | MAE | MSE | MAE |
| Traffic | 96 | **0.486** | **0.259** | 0.550 | 0.363 | 0.575 | 0.376 |
| | 192 | **0.525** | **0.284** | 0.536 | 0.343 | 0.557 | 0.353 |
| | 336 | **0.594** | **0.298** | 0.642 | 0.429 | 0.697 | 0.453 |
| | 720 | 0.686 | **0.327** | **0.675** | 0.444 | 0.727 | 0.397 |
| | Avg | **0.573** | **0.292** | 0.601 | 0.395 | 0.639 | 0.395 |
| Illness | 24 | **4.071** | **1.482** | 4.221 | 1.525 | 5.189 | 1.572 |
| | 36 | **3.962** | **1.443** | 4.235 | 1.496 | 4.676 | 1.530 |
| | 48 | **4.006** | **1.477** | 4.349 | 1.515 | 5.060 | 1.642 |
| | 60 | **4.053** | **1.492** | 4.632 | 1.559 | 4.667 | 1.603 |
| | Avg | **4.023** | **1.474** | 4.359 | 1.524 | 4.898 | 1.587 |

more profound understanding of time series background knowledge. The instruction part has two task instructions to guide the model in understanding the time series data and generating predictions. We compared the impact on performance when these two parts varied separately, as shown in Table 6. However, when their expression changes but the basic meaning remains consistent, modifying the domain description and instructions does not significantly affect the model's performance. The detailed modification of the prompt can be found in Appendix B.3.

### 4.4. Zero-Shot Transferability Analysis

In this section, we conduct an in-depth analysis of the zero-shot performance of our pre-trained model compared to other cross-domain training models in unseen domains. Table 7 demonstrates that LangTime consistently outperforms baseline models in most cases. This highlights the effectiveness of TCPs in enhancing the model's adaptability to diverse, unseen time series distributions. Additionally, we further analyze LangTime's cross-domain transferability in Appendix C.1.

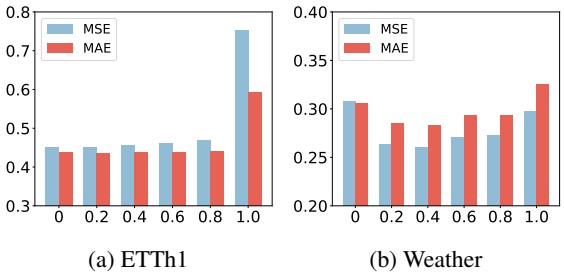

|   (a) ETTh1   |   (b) Weather   |

*Figure 4.* Parameter sensitivity analysis on $\alpha$ used in Equation (1).

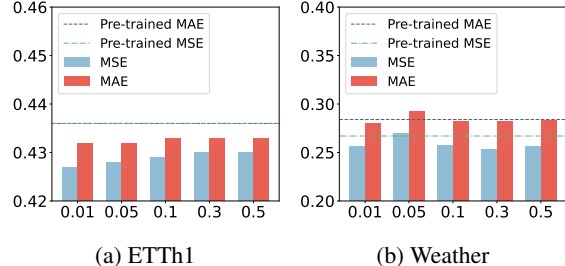

|   (a) ETTh1   |   (b) Weather   |

*Figure 5.* Parameter sensitivity analysis on $\tau$ used in Equation (5).

### 4.5. Parameter Analysis

**The Sensitivity of $\alpha$.** Figure 4 presents the impact of $\alpha$ on model performance. When only the prediction task ($\alpha = 0$) or only the reconstruction task ($\alpha = 1$) is applied, the model exhibits suboptimal performance. However, when balancing both tasks ($\alpha = 0.4$ or $\alpha = 0.6$), LangTime achieves the best results across both datasets. This highlights the importance of the reconstruction task in guiding the LLM's understanding of time series structures.

**The Sensitivity of $\tau$.** Figure 5 illustrates the impact of $\tau$ on TimePPO's rewards function. Since $\tau$ influences the distribution of reward scores, its effect varies based on the dataset's MSE and MAE convergence values. For ETTh1, performance remains relatively stable across different values of $\tau$, indicating low sensitivity. In contrast, for Weather, an improper $\tau$ selection leads to noticeable fluctuations, occasionally causing performance degradation after fine-tuning. Nevertheless, in most cases, TimePPO exhibits robustness with respect to $\tau$ within a certain range.

**The Sensitivity of $\beta$ and $\eta$.** Figure 6 illustrates the model's sensitivity to the parameters $\beta$ and $\eta$. Here, $\beta$ controls the MSE penalty term, aiming to regulate the extent of policy updates from the reward score perspective. For $\eta$, we referenced the alignment tax design in InstructGPT (Ouyang et al., 2022), enhancing the model's prediction capability and training stability. The sensitivity analysis experiments revealed no significant impact of these parameters, demonstrating the robustness of our proposed method.

### 4.6. Model Analysis

**Error Accumulation Analysis.** To evaluate how TimePPO improves model resilience against cumulative errors, we

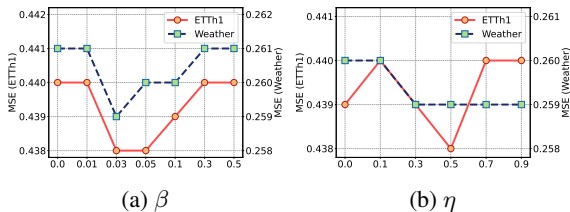

*Figure 6.* Parameter sensitivity analysis on $\beta$ and $\eta$.

*Table 8.* Comparison of algorithms in mitigating accumulated error on ETTh1 and Weather datasets. Reporting last-step metrics for long-term autoregressive forecasting (336, 720), with last 96 points for 336 and 48 points for 720.

| Dataset | | Pre-Training | | SFT | | TimePPO | |
|---|---|---|---|---|---|---|---|
| | | MSE | MAE | MSE | MAE | MSE | MAE |
| ETTh1 | 336 | 0.527 | 0.469 | 0.526 | 0.468 | **0.519** | **0.464** |
| | 720 | 0.541 | 0.513 | 0.539 | **0.512** | **0.537** | 0.513 |
| | Avg | 0.534 | 0.491 | 0.533 | 0.490 | **0.528** | **0.489** |
| Weather | 336 | 0.364 | 0.353 | 0.364 | 0.353 | **0.358** | **0.350** |
| | 720 | 0.467 | 0.418 | 0.475 | 0.418 | **0.462** | **0.412** |
| | Avg | 0.415 | 0.385 | 0.419 | 0.386 | **0.410** | **0.381** |

analyzed prediction metrics at the final step of long-term forecasting (336 and 720). As shown in Table 8, TimePPO demonstrates superior performance in these critical final-step predictions where accumulated errors have greatest impact. This improvement stems from TimePPO's unique approach: its value function estimates sequence-wide returns while advantage calculations assess each step's long-term value relative to these estimates. Unlike methods that focus solely on immediate next-step prediction, these mechanisms optimize the entire sequence's prediction quality. Through this holistic optimization, TimePPO effectively mitigates the cumulative error problem inherent in autoregressive models.

**T-SNE Visualization.** In this part, we focus on analyzing the three channels of the ETTh1 and Weather datasets. The ETTh1 dataset emphasizes energy-related features, including *High Useful Load*, *High Useless Load*, and *Middle Useful Load*, while the Weather dataset comprises features related to the natural environment, such as *Air Temperature*, *Air Pressure*, and *Potential Temperature*. We employ T-SNE visualization techniques to analyze LangTime's performance in processing these features, aiming to better understand the model's adaptability across different domains.

As shown in Figure 7, initially, TE processed all channels uniformly, leading to overlaps in representations between different domains such as *High Useful Load* and *High Useless Load* within the ETTh1 dataset. However, TCPs provided channel guidance to LLMs, enabling them to effectively distinguish these features and uncover their semantic relationships. Specifically, in the Weather dataset, *Air Temperature* and *Potential Temperature* exhibit stronger correlation compared to *Air Pressure*, demonstrating the role of language guidance in uncovering intrinsic dependencies between different channels.

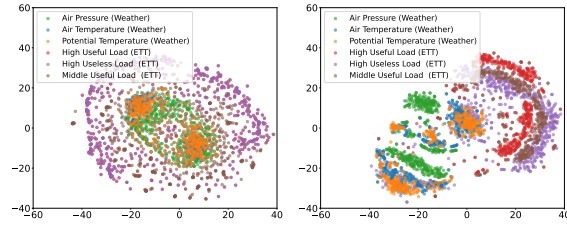

(a) Temporal Representations    (b) Compressed Token

*Figure 7.* T-SNE visualization of the mean pooling of temporal representations generated by Temporal Encoder and compressed token generated by LLM.

The information of given time series:
Period: 2020-12-19 05:40:00 to 2020-12-19 21:30:00,
Dataset: Meteorological indicator data with ten minute sample rate,
Channel: Air Pressure,
Value: [ENC][ENC][ENC][ENC].
Please compress this series into one word: <|EMB|>.
Base on the given information, predict next 4 values:

*Figure 8.* Attention map visualization for the last token, which is the actual prediction token preceding <|OUT|>. Special tokens and system prompts are hidden, and softmax values are recalculated over the remaining tokens to provide a clearer representation.

**Attention Map Analysis.** To further examine how LLMs interpret and integrate time series representations, we visualize the last token attention weights from the final layer of the LLM in LangTime that is ultimately processed through the MLP to generate predictions, as shown in Figure 8. The results indicate that: ① The last token focuses primarily on the time series representations and the token preceding the compressed token placeholder, which plays a key role in reconstructing the input sequence. ② LLMs also attend to contextual and prediction-relevant information, reinforcing the effectiveness of TCPs in aligning time series with LLMs. These findings confirm that TCPs enhance LLMs' ability to differentiate time series patterns, preserve domain-specific knowledge, and capture meaningful temporal dependencies.

## 5. Conclusion

In this work, we propose LangTime, a novel generalized model designed to address the challenges of leveraging multi-domain datasets for improving downstream time series forecasting. By incorporating TCPs, LangTime enhances the LLM's ability to interpret time series embeddings and generate accurate predictions. Additionally, we introduce TimePPO, a fine-tuning algorithm specifically designed for time series to effectively mitigate error accumulation in long-term forecasting. Extensive experiments demonstrate that LangTime achieves state-of-the-art performance on standard benchmarks and exhibits strong zero-shot transferability to unseen domains. Future work will concentrate on broadening LangTime's applicability to encompass a wider array of time series analysis tasks.

## Acknowledgements

This work is supported by the National Natural Science Foundation of China under Grant 62376194, and in part by China Scholarship Council Grant 202406250137.

## Impact Statement

The objective of this study is to propel the advancement of large models within the realm of time series analysis. In this work, we have developed a unified time series prediction model grounded in large language models and have implemented specific enhancements tailored for time series using the proximal policy optimization algorithm. The optimized model exhibits significant predictive performance across multiple real-world datasets. The algorithm we propose plays a pivotal role in mitigating the impact of cumulative errors in autoregressive time series prediction models. This study provides valuable insights for future related research and offers practical application value for practitioners. Our research is primarily focused on scientific exploration and does not present any discernible negative social impacts.

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

# A. Method Design Details

## A.1. Pre-Training Loss Details

During pre-training, we combine two tasks, as shown in Equation (1). For each task, we use the Huber loss, as illustrated in Equation (9) and Equation (10).

$$\mathcal{L}_{\text{reconstruction}} = \begin{cases} \frac{1}{2}(x - \hat{x})^2, & \text{if } |x - \hat{x}| \leq \delta, \\ \delta \cdot (|x - \hat{x}| - \frac{1}{2}\delta), & \text{otherwise.} \end{cases} \tag{9}$$

$$\mathcal{L}_{\text{prediction}} = \begin{cases} \frac{1}{2}(y - \hat{y})^2, & \text{if } |y - \hat{y}| \leq \delta, \\ \delta \cdot (|y - \hat{y}| - \frac{1}{2}\delta), & \text{otherwise.} \end{cases} \tag{10}$$

## A.2. Temporal Comprehension Prompts Details

Table 9 provides a description of each channel in the datasets used in TCPs. The full prompts used in TCPs are as follows.

---

**Temporal Comprehension Prompts**

<|im_start|>system
You are a helpful assistant, and your target is to summarize a time series and predict the next time series. Note: Value means the actual values of the time series, where each token represents data for <PATCH SIZE> consecutive time points. <|im_end|>
<|im_start|>user
The information of the given time series:
Period: <Timestamp>,
Dataset: <Dataset Information>,
Channel: <Channel Information>,
Value: <Time Series Representation>,
Please compress this series into one word: <|EMB|>.
Based on the given information, predict next <N> values: <|im_end|>
<|im_start|>assistant
<|OUT|><|im_end|>

---

In TCPs, <Timestamp> , <Dataset Information>, and <Channel Information> define the domain description, encoding dataset-specific characteristics to help the model incorporate relevant contextual information. <Time Series Representation> , generated by the Temporal Encoder, represents the processed time series features. <|EMB|> serves as a placeholder for the temporal series embeddings. <N> specifies the number of patches predicted concurrently and <|OUT|> represents the placeholder for the predicted outputs.

## A.3. Rewards Dimension Details

The Rewards Function we propose assesses the significance of the output across three distinct dimensions. As delineated in Table 10, we independently compute the Mean Squared Error (MSE), Mean Absolute Error (MAE), and KL divergence between the predicted outcomes and the ground truth. For KL divergence, we first compute the mean and variance based on the two sequences, and then calculate the KL divergence between the two continuous distributions. Subsequently, their reciprocals are employed as evaluation metrics. Ultimately, the reward score for the predicted outcomes is determined via Equation (5). The proximity of the predicted result $\hat{y}_t$ to the ground truth $y_t$ is directly proportional to the reward score attained.

## A.4. TimePPO Details

Unlike the discrete action space in NLP, the output of time series prediction models is a sequence in a continuous space. Therefore, we have redesigned the loss function based on NLP to measure the prediction results of the model from the probability distribution in the continuous space corresponding to the time series.

TimePPO is specifically designed to handle the continuous action space of time series forecasting, where the ratio of predicted mean probabilities is computed. This design prevents instability arising from excessive policy updates, thereby

*Table 9.* Dataset Information and Channel Information used in TCPs. *i* means column index in dataset *Traffic* and *Electricity*. The *Traffic* and *Illness* datasets are utilized to evaluate Zero-Shot performance, whereas the remaining datasets are employed for both training and testing.

| Dataset | Description | Channel Description |
|---|---|---|
| ETT | An hourly-sampled (minutely-sampled in ETTm1 and ETTm2) electricity transformer dataset intended for electrical asset monitoring, collected from one area in a province in China. | High Useful Load
High Useless Load
Middle Useful Load
Middle Useless Load
Low Useful Load
Low Useless Load
Oil Temperature |
| Weather | Meteorological indicator data with ten minute sample rate. | Air Pressure
Air Temperature
Potential Temperature
Dew Point Temperature
Relative Humidity
Saturation Water Vapor Pressure
Actual Water Vapor Pressure
Water Vapor Pressure Deficit
Specific Humidity
Water Vapor Concentration
Air Density
Wind Velocity
Maximum Wind Velocity
Wind Direction
Precipitation
Duration of Precipitation
Short Wave Downward Radiation
Photosynthetically Active Radiation
Maximum Photosynthetically Active Radiation
Internal Logger Temperature
$CO_2$ Concentration of Ambient Air |
| Exchange | Daily exchange rates of the US dollar to eight different currencies ranging from 1990 to 2016. | Exchange rate of the US dollar to the Australian dollar
Exchange rate of the US dollar to the British pound
Exchange rate of the US dollar to the Canadian dollar
Exchange rate of the US dollar to the Swiss franc
Exchange rate of the US dollar to the Chinese yuan
Exchange rate of the US dollar to the Japanese yen
Exchange rate of the US dollar to the New Zealand dollar
Exchange rate of the US dollar to the Singapore dollar |
| Electricity | Hourly electricity consumption of 321 customers from 2012 to 2014. | Electricity consumption of customer $i$ |
| Traffic | Hourly road occupancy rates data from 862 detectors across the freeways of the San Francisco Bay area, covering the years 2015 to 2016. | Road occupancy rates detected by detector $i$ |
| Illness | Weekly influenza-like illness (ILI) patient data from US Centers for Disease Control (2002-2021) showing ILI patient ratio. | Weight ILI Rate
Unweight ILI Rate
Number of ILI patients aged between 0-4 years old
Number of ILI patients aged between 5-24 years old
Total number of ILI patients across all age groups
Number of sentinel providers
Total patients |

*Table 10.* Dimensions of Rewards Function

| Dimension | Calculation Formula |
|---|---|
| MSE | $\mathcal{R}_{\text{MSE}} = \frac{1}{\frac{1}{n}\sum_{t=1}^{n}(\hat{y}_t - y_t)^2}$ |
| MAE | $\mathcal{R}_{\text{MAE}} = \frac{1}{\frac{1}{n}\sum_{t=1}^{n}|\hat{y}_t - y_t|}$ |
| KL Divergence | $\mathcal{R}_{\text{KL}} = \frac{1}{D_{\text{KL}}(P\|Q)}$ |

enhancing robustness in long-term forecasting.

We consider time series prediction as a continuous action space and calculate the ratio of predicted mean probabilities:

$$\pi(a|y) = \frac{1}{\sqrt{2\pi\sigma^2(y)}}\exp\left(-\frac{(a-\mu(y))^2}{2\sigma^2(y)}\right), \tag{11}$$

where $\mu(y)$ represents the mean of sequence $y$, while $\sigma^2(y)$ denotes the variance of sequence $y$. Based on Equation (11), we employ the probability policy ratio $r(\theta)$ to quantify the variations between the two policies.

$$r(\theta) = \frac{\pi_\theta(\mu(y_{\text{new}})|y_{\text{new}})}{\pi_{\theta_{\text{old}}}(\mu(y_{\text{new}})|y_{\text{old}})}. \tag{12}$$

Based on this design, $r(\theta)$ can accurately reflect the impact on the distribution of prediction results before and after the model parameter updates. Compared with directly calculating the numerical error of prediction results, it exhibits stronger robustness.

## B. Experimental Details

### B.1. Datasets

We perform comprehensive experiments on seven extensively employed time series datasets for long-term forecasting. Adhering to the data processing and train-validation-test set split protocol established in TimesNet (Wu et al., 2022), we ensure that the train, validation, and test datasets are rigorously partitioned in chronological order to prevent data leakage. Regarding the long-term forecasting configurations, we set the context length of LangTime and the lookback length of other comparative methods to 96, while the forecast length varies among {96, 192, 336, 720}. Brief descriptions of the pre-training datasets are as follows: (1) **ETT** (Zhou et al., 2021) includes data for monitoring electricity transformers from July 2016 to July 2018, comprising four subsets: ETTm1, ETTm2, ETTh1, and ETTh2. (2) **Electricity** contains hourly power consumption data for 321 clients from 2012 to 2014. (3) **Exchange** (Lai et al., 2018) records daily exchange rates for eight countries from 1990 to 2016. (4) **Weather** is recorded every 10 minutes in 2020, featuring 21 meteorological indicators such as temperature, humidity, and precipitation. Additionally, we evaluated the zero-shot performance of LangTime using two datasets from different domains: (1) **Illness** includes weekly data on patients with seven influenza-like illnesses from 2002 to 2021. (2) **Traffic** includes data on hourly road occupancy rates, gathered by 862 detectors across the freeways of the San Francisco Bay area from 2015 to 2016. Owing to the constraints of computational resources and given our adoption of a channel-independent approach, we partition the number of channels in certain datasets. Table 11 illustrates the number of channels present in each sub-dataset. This strategy facilitates more efficient management of batch size during cross-domain training.

### B.2. Experimental Setting

LangTime was implemented using PyTorch (Paszke et al., 2019), and all experiments were executed on 8 NVIDIA A100 80GB GPUs. We employed the AdamW optimizer (Loshchilov, 2017) with an initial learning rate of $1 \times 10^{-4}$. A cosine annealing schedule with warmup was utilized for learning rate decay during pre-training, with a warmup rate set at 0.05. The Temporal Encoder in LangTime comprises 4 layers utilizing group query attention (Ainslie et al., 2023), with a query head count of 8, and key and value head counts of 2. The dimension of the latent space is set to 512. The patching process employs a patch size of 24, and LangTime predicts 4 patches simultaneously.

*Table 11.* Summary of datasets used for pre-training.

| Dataset Name | Channels | Frequency | Time Points | Channel Split | Application Domain |
|---|---|---|---|---|---|
| ETTm1/ETTm2 | 7 | 15 mins | 57,507 | - | Energy Infrastructure Observation |
| ETTh1/ETTh2 | 7 | 1 hour | 14,307 | - | Energy Infrastructure Observation |
| Electricity | 321 | 1 hour | 26,211 | 7 | Electricity Consumption |
| Weather | 21 | 10 mins | 52,603 | 7 | Meteorologic Monitoring |
| Exchange | 8 | 1 day | 7,207 | - | Foreign Exchange Market |

**Pre-training.** During the warmup phase, we established $\alpha = 0.7$ to facilitate LLMs in comprehending the time series, subsequently reducing $\alpha$ to 0.5 during the learning rate decay phase. The Mask Rate was set at 0.4, and for the pre-trained model with a lookback length of 96, we designated the input lengths as $\{96, 288, 480, 672\}$. The batch size is set to 48 in pre-training.

**Fine-tuning.** Customizing the fine-tuning parameters can yield optimal results for different datasets. For instance, in the Weather dataset, we configure the initial learning rate to $1 \times 10^{-6}$, maintain the lookback window at 96, and execute predictions with target lengths of 672, with a clip range $\epsilon$ set to 0.1. The parameter $\tau$ is set to 0.1, while $\xi$ is set to 0.9.

### B.3. Ablation Study Details

In Table 6, we compare the impact of modifying the Instruction part and the Background part in TCPs on model performance. For the Instruction part, it is modified as follows:

---

**Temporal Comprehension Prompts (Modified)**

The details of the provided time series:
Period: <Timestamp>,
Dataset: <Dataset Information>,
Channel: <Channel Information>,
Value: <Time Series Representation>,
Please summarize this series in a single term: <|EMB|>.
Using the given details, forecast the upcoming <N> values: <|OUT|>.

---

In Table 6, we also made modifications to the descriptions of the ETTh1 and Weather datasets. The specific modification details are shown in Table 12.

*Table 12.* Details of Modifying the Background Description Prompt for the ETTh1 and Weather Datasets

| Prompt | Background of ETTh1 | Background of Weather |
|---|---|---|
| Original | An hourly-sampled electricity transformer dataset intended for electrical asset monitoring, collected from one area in a province in China. | Meteorological indicator data with ten minute sample rate. |
| Modified | An hourly-sampled dataset of electricity transformers designed for monitoring electrical assets. | Data on meteorological indicators sampled every ten minutes. |

## C. Supplementary Results

### C.1. Cross-Domain Transfer Capability Analysis

**Zero-Shot Transferability Analysis.** To further investigate the transferability of LangTime across different domains, we pre-train LangTime on the ETTh1 and ETTm1. Following the experimental setup of UniTime(Liu et al., 2024d), we evaluate the model on three datasets: ETTh2 (which belongs to the same domain as the source), Electricity (a different domain with certain underlying similarities to the source), and Weather (which represents a completely unrelated domain). The results are summarized in Table 13. Compared to UniTime, which also employs natural language for enhancement, our model achieves

superior performance on most datasets. This demonstrates the effectiveness of our proposed language-guided approach in enhancing the ability of large language models to understand time series data.

*Table 13.* Zero-shot transferability comparisons. Models trained and validated under uniform data settings

| Method | | ETTm2 | | Weather | | Electricity | |
|---|---|---|---|---|---|---|---|
| | | MSE | MAE | MSE | MAE | MSE | MAE |
| LangTime | 96 | **0.199** | **0.271** | **0.240** | **0.279** | **0.319** | **0.397** |
| | 192 | **0.258** | **0.310** | 0.294 | 0.321 | **0.339** | **0.418** |
| | 336 | **0.322** | **0.350** | **0.337** | **0.346** | **0.384** | **0.451** |
| | 720 | **0.425** | **0.409** | **0.411** | 0.393 | **0.467** | **0.503** |
| | Avg | **0.301** | **0.335** | **0.320** | **0.335** | **0.377** | **0.442** |
| UniTime | 96 | 0.207 | 0.284 | 0.244 | 0.281 | 0.432 | 0.505 |
| | 192 | 0.267 | 0.320 | **0.293** | **0.316** | 0.453 | 0.525 |
| | 336 | 0.325 | 0.357 | 0.342 | 0.347 | 0.459 | 0.532 |
| | 720 | 0.426 | 0.413 | 0.414 | **0.391** | 0.486 | 0.552 |
| | Avg | 0.306 | 0.343 | 0.323 | 0.334 | 0.458 | 0.529 |

**Comparison with Time Series Foundation Model.** Foundation models for time series are typically pre-trained on large-scale time series datasets and exhibit strong domain transfer capabilities (Ansari et al., 2024; Liu et al., 2024f). To further assess the impact of the language-guided approach on domain transferability, we compare LangTime, pre-trained on three datasets (Weather, ECL, and Exchange), with TimesFM(Das et al., 2024), which is pre-trained on a large-scale dataset. The results are presented in Table 14. To ensure the invisibility of test data, we conduct evaluations on four datasets: ETTh1, ETTh2, ETTm1, and ETTm2. Even when pre-trained on fewer datasets, LangTime still demonstrates competitive performance and achieves favorable results on multiple datasets. This further validates the effectiveness of our proposed framework.

*Table 14.* Zero-shot performance comparison of LangTime and foundation model

| Method | | ETTh1 | | ETTh2 | | ETTm1 | | ETTm2 | |
|---|---|---|---|---|---|---|---|---|---|
| | | MSE | MAE | MSE | MAE | MSE | MAE | MSE | MAE |
| LangTime | 96 | **0.493** | **0.449** | **0.331** | 0.374 | 0.893 | 0.590 | **0.216** | 0.303 |
| | 192 | 0.536 | 0.475 | **0.416** | **0.423** | 0.866 | 0.595 | **0.257** | **0.324** |
| | 336 | 0.570 | 0.495 | **0.444** | **0.447** | 0.920 | 0.621 | **0.342** | **0.375** |
| | 720 | **0.547** | **0.504** | **0.474** | **0.473** | 0.951 | 0.653 | 0.448 | **0.429** |
| | Avg | **0.537** | **0.481** | **0.416** | **0.429** | 0.907 | 0.615 | **0.316** | **0.358** |
| TimesFM | 96 | 0.516 | 0.429 | 0.368 | **0.358** | **0.687** | **0.490** | 0.276 | **0.300** |
| | 192 | **0.533** | **0.461** | 0.461 | 0.434 | **0.826** | **0.565** | 0.416 | 0.365 |
| | 336 | **0.560** | **0.467** | 0.507 | 0.448 | **0.780** | **0.569** | 0.556 | 0.443 |
| | 720 | 1.075 | 0.652 | 0.549 | 0.503 | **0.862** | **0.620** | **0.440** | 0.434 |
| | Avg | 0.671 | 0.502 | 0.471 | 0.436 | **0.789** | **0.561** | 0.422 | 0.386 |

### C.2. TimePPO Fine-Tuning Scalability Analysis

**Impact of Fine-Tuning Data Volume.** We analyzed the impact of the amount of data used for fine-tuning on model performance, as shown in Table 15. Models pre-trained across multiple domains exhibit performance improvements after fine-tuning with target domain datasets, which supports the effectiveness of our proposed method. Furthermore, even with limited data, the models fine-tuned using TimePPO still achieve favorable performance, enhancing their scalability in resource-constrained environments.

*Table 15.* Performance Impact of Fine-Tuning Data Volume on ETTh1 and Weather Datasets. Representing Data Usage at 5%, 10%, 15%, and 20%

| Dataset | | Pre-training | | 5% | | 10% | | 15% | | 20% | |
|---|---|---|---|---|---|---|---|---|---|---|---|
| | | MSE | MAE | MSE | MAE | MSE | MAE | MSE | MAE | MSE | MAE |
| ETTh1 | 96 | 0.388 | 0.396 | 0.385 | 0.397 | 0.384 | 0.397 | **0.383** | **0.396** | **0.383** | **0.396** |
| | 192 | 0.435 | 0.425 | 0.433 | 0.426 | 0.433 | 0.425 | 0.431 | 0.424 | **0.430** | **0.424** |
| | 336 | 0.469 | 0.444 | 0.467 | 0.444 | 0.469 | 0.443 | **0.467** | **0.442** | **0.467** | 0.443 |
| | 720 | 0.466 | 0.462 | 0.464 | 0.459 | 0.462 | **0.458** | **0.460** | **0.458** | **0.460** | 0.459 |
| | Avg | 0.439 | 0.432 | 0.437 | 0.432 | 0.437 | 0.431 | **0.435** | **0.430** | **0.435** | **0.430** |
| Weather | 96 | 0.169 | 0.205 | **0.162** | 0.199 | **0.162** | 0.200 | **0.162** | 0.202 | **0.162** | **0.198** |
| | 192 | 0.228 | 0.262 | 0.225 | 0.256 | 0.225 | 0.257 | 0.224 | 0.256 | **0.223** | **0.255** |
| | 336 | 0.278 | 0.297 | 0.278 | 0.297 | 0.277 | 0.298 | **0.275** | **0.297** | **0.275** | 0.298 |
| | 720 | 0.360 | 0.354 | 0.350 | 0.345 | 0.349 | 0.345 | 0.347 | **0.344** | **0.346** | 0.345 |
| | Avg | 0.259 | 0.280 | 0.254 | 0.274 | 0.253 | 0.275 | 0.252 | 0.275 | **0.252** | **0.274** |

**Impact of Fine-tuning Different Components.** We compare the effects of freezing certain components during TimePPO fine-tuning, and the results are shown in Table 16. Full-parameter fine-tuning of LangTime achieves the best performance, while fine-tuning only the LLM component yields results close to those of full-parameter fine-tuning. Notably, even when only the TE component is fine-tuned (with less than 3% of the total parameters), the model still achieves performance comparable to fine-tuning a larger proportion of parameters. This further demonstrates the adaptability of our method under resource-constrained settings.

*Table 16.* Effects of Fine-tuning Different Components of LangTime

| Dataset | | Pre-training | | Full | | LLM | | TE | |
|---|---|---|---|---|---|---|---|---|---|
| | | MSE | MAE | MSE | MAE | MSE | MAE | MSE | MAE |
| ETTh1 | 96 | 0.388 | **0.396** | **0.383** | **0.396** | **0.383** | **0.396** | 0.384 | **0.396** |
| | 192 | 0.435 | 0.425 | **0.430** | **0.424** | **0.430** | **0.424** | 0.431 | 0.425 |
| | 336 | 0.469 | 0.444 | **0.467** | **0.443** | **0.467** | **0.443** | 0.468 | 0.445 |
| | 720 | 0.466 | 0.462 | **0.460** | **0.459** | **0.460** | **0.459** | 0.465 | 0.461 |
| | Avg | 0.439 | 0.432 | **0.435** | **0.431** | **0.435** | **0.431** | 0.437 | 0.432 |
| Weather | 96 | 0.169 | 0.205 | **0.162** | **0.199** | **0.162** | **0.199** | 0.163 | **0.199** |
| | 192 | 0.228 | 0.262 | **0.225** | **0.256** | **0.225** | **0.256** | 0.226 | **0.256** |
| | 336 | **0.278** | **0.297** | 0.278 | 0.298 | **0.278** | **0.297** | **0.278** | **0.297** |
| | 720 | 0.360 | 0.354 | **0.350** | **0.345** | **0.350** | **0.345** | 0.352 | **0.345** |
| | Avg | 0.259 | 0.280 | **0.254** | **0.274** | **0.254** | **0.274** | 0.255 | **0.274** |

## C.3. More Parameter Sensitivity Analysis

We tested the impact of mask rate and single-step prediction length on LanTime during pre-training on the ETTh1 dataset, with results shown in Table 17. Among the three output lengths, the model performed best when the output length was 96. This is because a shorter output length necessitates more steps to complete the prediction, leading to greater cumulative error. On the other hand, while a longer single-step prediction length reduces the number of steps, it results in decreased accuracy in single-step predictions.

The mask rate balances two objectives: (1) preventing overfitting in datasets with varying convergence speeds, and (2) enhancing temporal pattern learning through reconstruction tasks. Table 17 shows that extreme mask rates (either too low or too high) degrade performance: low rates fail to improve temporal pattern extraction, while high rates impair long-horizon

predictions.

*Table 17.* Sensitivity analysis of mask rate and output sequence length on the ETTh1 dataset. **Bold**: best performance for the same output length, Underline: the best performance for the same mask rate.

| Mask Rate | | 0 | | 0.2 | | 0.4 | | 0.6 | |
|---|---|---|---|---|---|---|---|---|---|
| | | MSE | MAE | MSE | MAE | MSE | MAE | MSE | MAE |
| 24 | 96 | 0.392 | 0.398 | **0.387** | **0.398** | 0.395 | 0.401 | 0.409 | 0.404 |
| | 192 | 0.456 | 0.433 | **0.442** | 0.434 | 0.449 | 0.432 | 0.456 | **0.431** |
| | 336 | 0.500 | 0.456 | **0.483** | 0.454 | 0.486 | 0.444 | **0.483** | **0.452** |
| | 720 | 0.517 | 0.481 | 0.492 | 0.465 | **0.487** | **0.464** | 0.497 | 0.466 |
| | Avg | 0.466 | 0.442 | **0.451** | 0.438 | 0.455 | **0.435** | 0.461 | 0.438 |
| 96 | 96 | 0.384 | 0.394 | **0.383** | 0.396 | **0.383** | **0.393** | **0.383** | 0.394 |
| | 192 | 0.437 | 0.424 | 0.436 | 0.425 | **0.434** | **0.423** | 0.438 | 0.424 |
| | 336 | 0.473 | **0.442** | **0.470** | 0.444 | **0.470** | 0.443 | 0.473 | 0.444 |
| | 720 | 0.484 | 0.462 | **0.473** | **0.462** | 0.475 | 0.463 | 0.482 | 0.466 |
| | Avg | 0.444 | 0.431 | 0.441 | 0.432 | **0.440** | **0.430** | 0.444 | 0.432 |
| 192 | 96 | 0.394 | 0.401 | **0.390** | **0.399** | 0.394 | 0.401 | **0.390** | **0.399** |
| | 192 | 0.450 | 0.429 | **0.445** | 0.426 | 0.450 | 0.429 | **0.445** | 0.428 |
| | 336 | 0.491 | 0.449 | 0.484 | **0.445** | 0.489 | 0.447 | **0.481** | 0.446 |
| | 720 | 0.503 | 0.468 | 0.493 | **0.462** | 0.499 | 0.464 | **0.487** | 0.465 |
| | Avg | 0.460 | 0.437 | 0.453 | **0.433** | 0.458 | 0.435 | **0.451** | 0.435 |

## C.4. Full Results

In this section, we present the experimental results that were not fully displayed in the main text. Specifically, Table 18 provides a detailed account of the impact of different fine-tuning algorithms on two models pre-trained across multiple domains. Table 19 and Table 20 display the complete results of the ablation experiments we conducted. Table 21 and Table 22 respectively display the experimental results for $\alpha$ and $\tau$ in the parameter sensitivity analysis.

## C.5. Cases

In this part, we visualize the forecasting results of LangTime. Figure 9 illustrates the performance comparison between pre-trained LangTime and LangTime fine-tuned via TimePPO under various settings. Figure 10 visualizes the zero-shot results of the pre-trained LangTime on the Traffic dataset. This demonstrates the exceptional performance of our proposed LangTime across domains, achieving impressive results even in unseen domains.

*Table 18.* Comparisons of forecasting performance among various fine-tuning algorithms.

| Method | | ETTm1 | | ETTm2 | | ETTh1 | | ETTh2 | | Electricity | | Exchange | | Weather | |
|---|---|---|---|---|---|---|---|---|---|---|---|---|---|---|---|
| | | MSE | MAE | MSE | MAE | MSE | MAE | MSE | MAE | MSE | MAE | MSE | MAE | MSE | MAE |
| LangTime$_{PT}$ | 96 | **0.323** | **0.346** | **0.184** | 0.258 | 0.394 | 0.395 | 0.301 | 0.334 | 0.199 | 0.277 | **0.086** | 0.205 | 0.184 | 0.203 |
| | 192 | 0.372 | 0.376 | **0.245** | 0.300 | 0.439 | 0.420 | 0.380 | 0.389 | 0.213 | 0.296 | 0.175 | 0.300 | 0.216 | 0.250 |
| | 336 | 0.419 | 0.403 | 0.308 | 0.339 | 0.464 | 0.442 | 0.412 | 0.419 | 0.234 | 0.316 | 0.329 | 0.412 | 0.275 | 0.293 |
| | 720 | 0.491 | 0.443 | 0.410 | 0.399 | 0.462 | 0.449 | 0.422 | 0.439 | 0.272 | 0.357 | 0.854 | 0.696 | 0.361 | 0.349 |
| | Avg | 0.401 | 0.392 | 0.287 | 0.324 | 0.440 | 0.427 | 0.379 | 0.395 | 0.230 | 0.312 | 0.361 | 0.403 | 0.259 | 0.274 |
| LangTime$_{SFT}$ | 96 | 0.324 | **0.346** | **0.184** | **0.257** | 0.400 | **0.388** | **0.286** | **0.322** | 0.186 | **0.264** | **0.086** | 0.205 | **0.170** | 0.206 |
| | 192 | **0.366** | **0.374** | **0.245** | **0.296** | 0.447 | **0.418** | 0.384 | 0.390 | 0.198 | 0.286 | 0.183 | 0.305 | 0.221 | 0.253 |
| | 336 | 0.417 | 0.403 | 0.304 | 0.337 | 0.478 | **0.437** | 0.416 | 0.423 | 0.210 | 0.293 | **0.325** | 0.412 | 0.284 | 0.299 |
| | 720 | 0.488 | **0.439** | 0.406 | 0.395 | 0.461 | 0.448 | 0.424 | 0.429 | 0.249 | 0.323 | 0.856 | 0.695 | 0.377 | 0.356 |
| | Avg | 0.399 | 0.391 | 0.285 | **0.321** | 0.447 | **0.423** | 0.378 | **0.391** | 0.211 | 0.291 | 0.362 | 0.404 | 0.263 | 0.279 |
| LangTime$_{TimePPO}$ | 96 | **0.319** | 0.348 | 0.188 | 0.258 | **0.391** | **0.388** | 0.299 | 0.336 | **0.181** | 0.266 | 0.089 | **0.201** | 0.178 | **0.202** |
| | 192 | 0.368 | 0.375 | **0.245** | 0.297 | **0.429** | 0.419 | **0.374** | **0.382** | **0.185** | **0.273** | **0.175** | **0.298** | **0.211** | **0.245** |
| | 336 | **0.413** | **0.402** | **0.301** | 0.336 | **0.462** | 0.440 | **0.410** | **0.418** | **0.198** | **0.281** | 0.329 | **0.409** | **0.269** | **0.286** |
| | 720 | **0.487** | **0.439** | **0.402** | 0.393 | **0.458** | **0.445** | **0.418** | **0.426** | **0.241** | **0.320** | **0.852** | **0.690** | **0.351** | **0.346** |
| | Avg | **0.397** | **0.391** | **0.284** | **0.321** | **0.435** | **0.423** | **0.375** | **0.391** | **0.201** | **0.285** | **0.361** | **0.400** | **0.252** | **0.270** |
| AutoTimes$_{PT}$ | 96 | **0.914** | **0.590** | 0.269 | 0.331 | 0.417 | 0.408 | 0.301 | 0.333 | 0.188 | 0.267 | 0.133 | 0.253 | 0.219 | **0.245** |
| | 192 | 0.966 | 0.616 | **0.326** | 0.364 | 0.484 | 0.444 | **0.410** | **0.398** | **0.217** | 0.291 | **0.253** | **0.357** | **0.298** | **0.310** |
| | 336 | **0.935** | 0.612 | **0.379** | **0.393** | 0.529 | 0.468 | 0.420 | 0.419 | 0.236 | 0.319 | 0.390 | **0.452** | **0.337** | 0.338 |
| | 720 | 0.954 | 0.630 | 0.473 | 0.442 | 0.549 | 0.494 | 0.439 | 0.444 | 0.272 | 0.346 | 0.931 | 0.730 | **0.415** | 0.383 |
| | Avg | 0.942 | 0.612 | 0.362 | 0.383 | 0.495 | 0.454 | 0.392 | **0.399** | **0.228** | 0.306 | 0.427 | **0.448** | 0.317 | 0.319 |
| AutoTimes$_{SFT}$ | 96 | 0.916 | 0.616 | 0.270 | 0.329 | **0.415** | **0.401** | 0.301 | 0.333 | **0.184** | **0.261** | **0.123** | **0.235** | 0.220 | **0.245** |
| | 192 | **0.956** | 0.605 | **0.326** | 0.365 | 0.480 | **0.438** | 0.432 | 0.416 | **0.217** | **0.284** | **0.253** | **0.357** | 0.299 | **0.310** |
| | 336 | 0.940 | 0.606 | 0.383 | **0.393** | 0.526 | **0.461** | 0.420 | 0.419 | 0.238 | 0.317 | 0.389 | **0.452** | 0.338 | **0.338** |
| | 720 | 0.969 | **0.626** | 0.477 | 0.444 | 0.544 | 0.493 | 0.448 | 0.444 | 0.274 | 0.367 | 0.932 | 0.732 | **0.415** | 0.383 |
| | Avg | 0.945 | 0.613 | 0.364 | **0.383** | 0.491 | 0.448 | 0.400 | 0.403 | **0.228** | 0.307 | 0.424 | 0.444 | 0.318 | 0.319 |
| AutoTimes$_{TimePPO}$ | 96 | **0.914** | 0.609 | 0.257 | 0.323 | 0.417 | 0.403 | **0.300** | **0.332** | 0.208 | 0.277 | 0.127 | 0.252 | **0.218** | **0.245** |
| | 192 | 0.961 | **0.598** | 0.333 | 0.365 | **0.476** | **0.438** | **0.410** | 0.406 | 0.221 | 0.286 | 0.256 | 0.360 | **0.298** | **0.310** |
| | 336 | 0.938 | **0.605** | 0.380 | 0.403 | **0.517** | 0.465 | **0.416** | **0.417** | 0.237 | **0.308** | **0.388** | 0.458 | **0.337** | **0.338** |
| | 720 | **0.948** | 0.627 | **0.471** | 0.443 | **0.528** | **0.479** | **0.436** | **0.439** | **0.267** | **0.345** | **0.928** | **0.727** | **0.415** | **0.378** |
| | Avg | **0.940** | **0.610** | **0.360** | **0.383** | **0.485** | **0.446** | **0.390** | **0.399** | 0.233 | **0.304** | **0.425** | 0.450 | **0.317** | **0.318** |

*Table 19.* Ablation studies on various components of temporal comprehension prompts on ETTh1 and Weather datasets.

| Language Guidance | Timestamp | Dataset Information | Channel Information | Length | ETTh1 | | Weather | |
|---|---|---|---|---|---|---|---|---|
| | | | | | MSE | MAE | MSE | MAE |
| ✓ | ✓ | ✓ | ✓ | 96 | 0.384 | 0.399 | 0.175 | 0.209 |
| | | | | 192 | 0.432 | 0.429 | 0.235 | 0.265 |
| | | | | 336 | 0.471 | 0.449 | 0.293 | 0.312 |
| | | | | 720 | 0.459 | 0.467 | 0.364 | 0.349 |
| | | | | Avg | 0.436 | 0.436 | 0.267 | 0.284 |
| ✓ | ✓ | ✓ | | 96 | 0.385 | 0.402 | 0.179 | 0.214 |
| | | | | 192 | 0.434 | 0.429 | 0.238 | 0.267 |
| | | | | 336 | 0.476 | 0.451 | 0.295 | 0.315 |
| | | | | 720 | 0.465 | 0.471 | 0.366 | 0.356 |
| | | | | Avg | 0.440 | 0.438 | 0.269 | 0.288 |
| ✓ | ✓ | | | 96 | 0.392 | 0.407 | 0.180 | 0.217 |
| | | | | 192 | 0.441 | 0.431 | 0.240 | 0.267 |
| | | | | 336 | 0.468 | 0.450 | 0.299 | 0.318 |
| | | | | 720 | 0.469 | 0.471 | 0.369 | 0.365 |
| | | | | Avg | 0.442 | 0.440 | 0.272 | 0.292 |
| ✓ | | | | 96 | 0.393 | 0.407 | 0.180 | 0.220 |
| | | | | 192 | 0.440 | 0.433 | 0.239 | 0.269 |
| | | | | 336 | 0.470 | 0.454 | 0.306 | 0.320 |
| | | | | 720 | 0.467 | 0.477 | 0.369 | 0.365 |
| | | | | Avg | 0.442 | 0.443 | 0.274 | 0.293 |

*Table 20.* Ablation studies on various dimensions of Rewards Function on ETTh1 and Weather datasets.

| Method | | ETTh1 | | Weather | |
|---|---|---|---|---|---|
| | | MSE | MAE | MSE | MAE |
| LangTime$_{PT}$ | 96 | 0.384 | 0.399 | 0.175 | 0.209 |
| | 192 | 0.432 | 0.429 | 0.235 | 0.265 |
| | 336 | 0.471 | 0.449 | 0.293 | 0.312 |
| | 720 | 0.459 | 0.467 | 0.364 | 0.349 |
| | Avg | **0.436** | **0.436** | **0.267** | **0.284** |
| All dimensions | 96 | **0.379** | **0.399** | **0.172** | **0.205** |
| | 192 | 0.426 | 0.427 | 0.226 | 0.263 |
| | 336 | 0.466 | 0.448 | 0.288 | 0.310 |
| | 720 | 0.445 | 0.458 | 0.352 | 0.342 |
| | Avg | **0.429** | **0.433** | **0.259** | **0.280** |
| TimePPO w/o $\mathcal{R}_{MSE}$ | 96 | 0.382 | 0.401 | 0.175 | 0.208 |
| | 192 | 0.430 | 0.429 | 0.229 | 0.264 |
| | 336 | 0.472 | 0.452 | 0.289 | 0.311 |
| | 720 | 0.456 | 0.465 | 0.359 | 0.342 |
| | Avg | **0.435** | **0.437** | **0.263** | **0.281** |
| TimePPO w/o $\mathcal{R}_{MAE}$ | 96 | 0.381 | 0.401 | **0.172** | 0.208 |
| | 192 | 0.429 | 0.430 | 0.225 | 0.264 |
| | 336 | 0.470 | 0.453 | 0.289 | 0.312 |
| | 720 | 0.453 | 0.468 | 0.357 | 0.345 |
| | Avg | **0.433** | **0.438** | **0.261** | **0.282** |
| TimePPO w/o $\mathcal{R}_{KL}$ | 96 | 0.382 | 0.401 | 0.176 | 0.207 |
| | 192 | 0.430 | 0.430 | 0.228 | 0.264 |
| | 336 | 0.472 | 0.454 | 0.289 | 0.310 |
| | 720 | 0.456 | 0.468 | 0.357 | 0.346 |
| | Avg | **0.435** | **0.438** | **0.262** | **0.282** |

*Table 21.* Full result of parameter sensitivity analysis on $\alpha$

| $\alpha$ | Length | ETTh1 | | Weather | | $\alpha$ | Length | ETTh1 | | Weather | |
|---|---|---|---|---|---|---|---|---|---|---|---|
| | | MSE | MAE | MSE | MAE | | | MSE | MAE | MSE | MAE |
| 0 | 96 | 0.389 | 0.399 | **0.165** | **0.204** | 0.2 | 96 | **0.388** | **0.395** | 0.182 | 0.225 |
| | 192 | 0.443 | 0.432 | 0.254 | 0.279 | | 192 | **0.442** | **0.429** | 0.234 | 0.267 |
| | 336 | 0.494 | 0.456 | 0.329 | 0.330 | | 336 | **0.491** | **0.452** | 0.284 | 0.302 |
| | 720 | 0.481 | 0.467 | 0.484 | 0.409 | | 720 | **0.480** | **0.466** | 0.355 | 0.348 |
| | Avg | 0.452 | 0.438 | 0.308 | 0.306 | | Avg | **0.450** | **0.435** | 0.264 | 0.285 |
| 0.4 | 96 | 0.394 | 0.400 | 0.176 | 0.220 | 0.6 | 96 | 0.399 | 0.402 | 0.185 | 0.229 |
| | 192 | 0.445 | 0.431 | **0.230** | **0.264** | | 192 | 0.449 | 0.432 | 0.244 | 0.276 |
| | 336 | 0.497 | 0.454 | **0.281** | **0.300** | | 336 | 0.503 | 0.454 | 0.293 | 0.309 |
| | 720 | 0.485 | 0.467 | **0.354** | **0.347** | | 720 | 0.499 | 0.468 | 0.364 | 0.357 |
| | Avg | 0.455 | 0.438 | **0.260** | **0.283** | | Avg | 0.462 | 0.439 | 0.271 | 0.293 |
| 0.8 | 96 | 0.406 | 0.406 | 0.192 | 0.235 | 1 | 96 | 0.768 | 0.583 | 0.224 | 0.276 |
| | 192 | 0.459 | 0.436 | 0.244 | 0.277 | | 192 | 0.757 | 0.587 | 0.273 | 0.311 |
| | 336 | 0.511 | 0.456 | 0.292 | 0.309 | | 336 | 0.754 | 0.595 | 0.315 | 0.336 |
| | 720 | 0.497 | 0.463 | 0.363 | 0.355 | | 720 | 0.730 | 0.603 | 0.379 | 0.376 |
| | Avg | 0.468 | 0.440 | 0.273 | 0.294 | | Avg | 0.752 | 0.592 | 0.298 | 0.325 |

*Table 22.* Full result of parameter sensitivity analysis on $\tau$

| $\tau$ | Length | ETTh1 | | Weather | | $\tau$ | Langth | ETTh1 | | Weather | |
|---|---|---|---|---|---|---|---|---|---|---|---|
| | | MSE | MAE | MSE | MAE | | | MSE | MAE | MSE | MAE |
| LangTime$_{PT}$ | 96 | 0.384 | 0.399 | **0.175** | **0.209** | 0.01 | 96 | **0.377** | **0.397** | **0.170** | 0.212 |
| | 192 | 0.432 | 0.429 | 0.235 | 0.265 | | 192 | **0.424** | **0.426** | 0.224 | 0.259 |
| | 336 | 0.471 | 0.449 | 0.293 | 0.312 | | 336 | **0.465** | **0.447** | 0.278 | 0.299 |
| | 720 | 0.459 | 0.467 | 0.364 | 0.349 | | 720 | **0.443** | **0.456** | 0.354 | 0.349 |
| | Avg | 0.436 | 0.436 | 0.267 | 0.284 | | Avg | **0.427** | **0.432** | 0.256 | 0.280 |
| 0.05 | 96 | 0.378 | 0.397 | 0.224 | 0.259 | 0.1 | 96 | 0.379 | 0.399 | 0.171 | 0.214 |
| | 192 | **0.424** | **0.426** | 0.224 | 0.259 | | 192 | 0.426 | 0.427 | 0.224 | 0.262 |
| | 336 | 0.466 | 0.448 | **0.278** | **0.299** | | 336 | 0.466 | 0.448 | 0.278 | 0.302 |
| | 720 | 0.445 | 0.458 | **0.354** | **0.349** | | 720 | 0.445 | 0.458 | 0.354 | 0.352 |
| | Avg | 0.428 | 0.432 | **0.270** | **0.292** | | Avg | 0.429 | 0.433 | 0.257 | 0.282 |
| 0.3 | 96 | 0.378 | 0.398 | 0.173 | 0.220 | 0.5 | 96 | 0.378 | 0.398 | 0.175 | 0.219 |
| | 192 | 0.425 | 0.427 | **0.221** | **0.262** | | 192 | 0.425 | 0.427 | 0.223 | 0.262 |
| | 336 | 0.467 | 0.449 | **0.271** | **0.298** | | 336 | 0.467 | 0.449 | 0.275 | 0.300 |
| | 720 | 0.448 | 0.460 | **0.348** | **0.348** | | 720 | 0.449 | 0.461 | 0.352 | 0.350 |
| | Avg | 0.430 | 0.433 | **0.253** | **0.282** | | Avg | 0.430 | 0.433 | 0.256 | 0.283 |

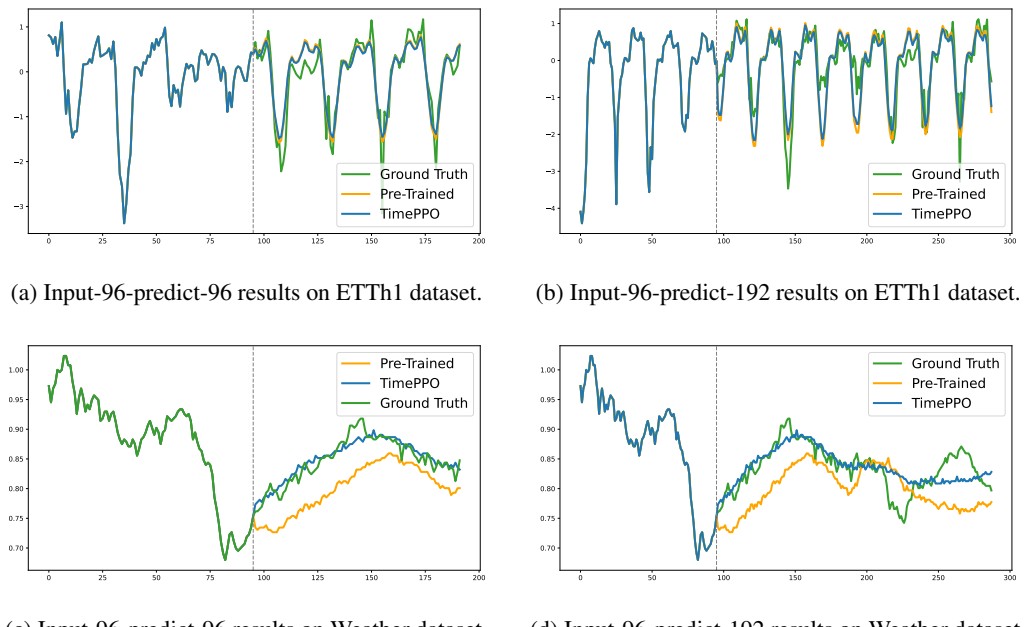

(a) Input-96-predict-96 results on ETTh1 dataset.

(b) Input-96-predict-192 results on ETTh1 dataset.

(c) Input-96-predict-96 results on Weather dataset.

(d) Input-96-predict-192 results on Weather dataset.

*Figure 9.* Long-term forecasting cases for ETTh1 and Weather dataset. Green lines are the ground truths, orange lines are the pre-trained model predictions and blue lines are the TimePPO fine-tuned model predictions. The vertical line indicates where the prediction starts.

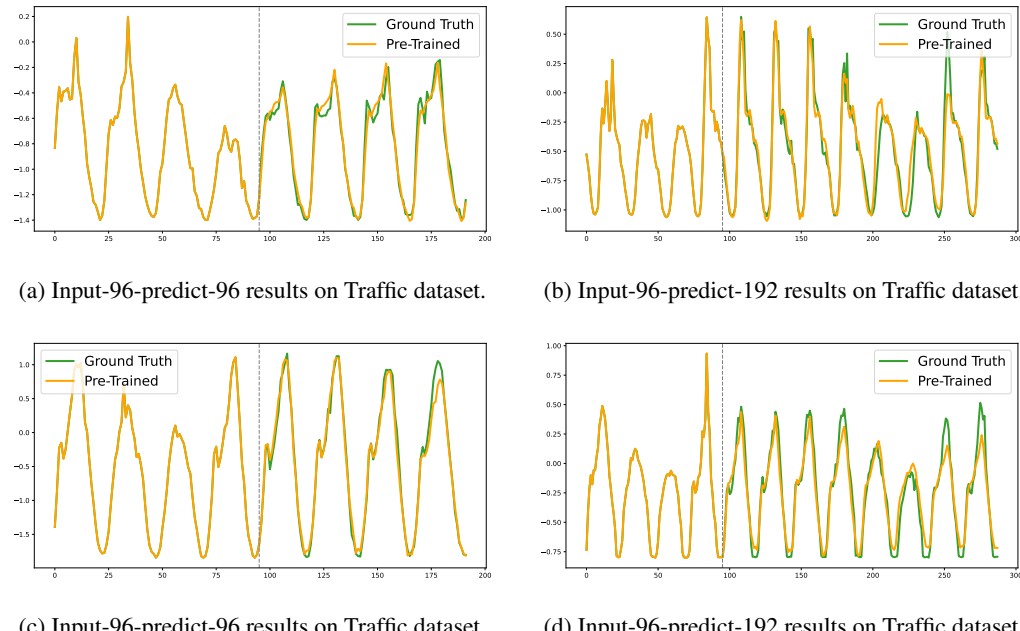

(a) Input-96-predict-96 results on Traffic dataset.

(b) Input-96-predict-192 results on Traffic dataset.

(c) Input-96-predict-96 results on Traffic dataset.

(d) Input-96-predict-192 results on Traffic dataset.

*Figure 10.* Zero-Shot forecasting cases for Traffic dataset. Green lines are the ground truths, orange lines are the pre-trained model predictions. The vertical line indicates where the prediction starts.

