# OpenReview forum: "LangTime: A Language-Guided Unified Model for Time Series Forecasting with Proximal Policy Optimization"
_ICML.cc/2025/Conference — ICML 2025 poster_

### Official Review · Reviewer_bFpj · 2025-03-06

**Overall Recommendation:** 4

**Summary:**

This paper introduces LangTime, which is a language guided unified model for time series forecasting. It integrates LLM with RL based fine-tuning using PPO for time series analysis. Among the designed components, the Temporal Comprehension Prompts (TCPs) align time series data with LLM by embedding data-specific and channel-specific instructions and the time series data is compressed into a single token to facilitate LLM's understanding. The proposed TimePPO addresses error accumulation in AR forecasting and introduces a multi-dimensional reward function for better predictions.  LangTime outperforms existing LLM-based time series models and shows strong. zero-shot transferability

**Claims And Evidence:**

Yes, the claims are clear and convincing.

**Essential References Not Discussed:**

The related work is complete.

**Experimental Designs Or Analyses:**

The experiments are sound and complete.

**Methods And Evaluation Criteria:**

Yes, the method and evaluation metrics are aligned.

**Other Comments Or Suggestions:**

Figure 2 contains too much compressed information, making it difficult to follow. Consider splitting it into two figures: one illustrating the overall pipeline and another focusing specifically on the TimePPO algorithm.

**Other Strengths And Weaknesses:**

Strength:

1. Integrating time series pre-training with PPO (Reinforcement Learning) is an innovative approach, offering a novel solution for aligning time series data with LLMs. In particular, TimePPO effectively mitigates accumulated errors in autoregressive forecasting.
2. The experiments are comprehensive and complete, showing the effectiveness of the proposed framework.
3. Present visualizations of compressed tokens to validate the effectiveness of the approach.

Weakness:

1. Do not mention the training efficiency, especially with autoregressive output approach.
2. On which datasets did you pre-train LangTime? The pre-training process is unclear to me, making it difficult to assess and potentially unfair to compare with zero-shot performance.

**Questions For Authors:**

1. Is there a specific reason for selecting Qwen2-0.5B-Instruction as the backbone LLM? I noticed that this choice differs from previous works.

**Relation To Broader Scientific Literature:**

It involves leveraging LLMs for time series analysis through RL-based pre-training, with the potential to serve as a foundation model for time series forecasting.

**Theoretical Claims:**

N/A

---

> ### Author Rebuttal · Authors · 2025-04-01
>
> We sincerely thank you for the thorough review and insightful comments. In our response, our model was jointly pre-trained on the ETTh1 and Weather datasets, and experiments involving the TimePPO stage were fine-tuned on individual datasets. We presented the average of the results, and detailed experimental results are available at [link](https://anonymous.4open.science/r/full-E4EE/README.md).
>
> >Suggestions: Figure 2 contains too much compressed information, making it difficult to follow.
>
> Thank you for your suggestion. We have re-drawn Figure 2 and presented it as Figure 1 in the [link](https://anonymous.4open.science/r/full-E4EE/README.md). In the next version, we will update some images in the paper for clearer expression.
>
> >W1: Do not mention the training efficiency, especially with autoregressive output approach.
>
> Our model achieved the results in the paper with just 1 epoch of pre-training, partially compensating for the slower training speed. Due to the complex procedure of the PPO algorithm, the TimePPO fine-tuning phase is slow. However, our experiments demonstrate that fine-tuning with limited data (Tab.6 in our response to reviewer **5obH**'s **Q4**) and mostly frozen parameters (Tab.2) still yields good performance, enhancing its scalability under resource constraints.
>
> Table 1 Training Speeds of Different Models
> ||Ours(pt)|UniTime|AutoTimes|TimeLLM
> |-|-|-|-|-|
> |Speed ms/iter|218|8|89|212
> |Parameter|0.5B|0.13B|7B|7B
>
> Table 2 Impact of Fine-tuning Data Volume on ETTh1
> |data rate|PT|5%|10%|15%|20%
> |-|-|-|-|-|-
> |MSE|0.439|0.437|0.437|0.435|**0.435**
> |MAE|0.432|0.432|0.431|0.430|**0.430**
>
> >W2: The pre-training process is unclear.
>
> We pre-trained on 7 datasets (ETT, Weather, Exchange, Electricity) and conducted zero-shot experiments on two unseen datasets (Traffic, Illness). All models in Table 5 in paper were trained under the same dataset settings, making the comparison fair.
>
> >Q1: Is there a specific reason for selecting Qwen2-0.5B-Instruction as the backbone LLM?
>
> Compared to GPT2, Qwen is pre-trained on larger datasets, offering better language understanding and stronger instruction-following abilities to better guide LLM in interpreting time series. While the common Llama-7b also has powerful capabilities, its large number of parameters led us to choose the smaller Qwen2-0.5B-Instruction model as the backbone. In Table x, we compared different backbones; GPT2's performance on Weather is close to Qwen, demonstrating our framework's effectiveness, but its performance on ETTh1 reveals its limitations.
>
> Table 3 Performance of Different Backbones
> |backbones|GPT2||Qwen||Linear||
> |-|-|-|-|-|-|-
> ||MSE|MAE|MSE|MAE|MSE|MAE
> |ETTh1|0.492|0.447|**0.448**|**0.431**|0.723|0.588
> |Weather|0.275|0.294|**0.270**|**0.277**|0.301|0.333

---

> > ### Comment · Reviewer_bFpj · 2025-04-04
> >
> > Thank you for addressing my questions. I will maintain my score, as the paper indeed presents a novel and promising framework to tackle the challenges in this area. One minor suggestion: during pretraining, it may be beneficial to consider using larger-scale datasets, such as those provided in Chronos or Moirai, to further enhance the model’s robustness and performance. This is an interesting and valuable contribution. Looking forward to the release of the pretrained model!

---

> > > ### Author Response · Authors · 2025-04-06
> > >
> > > Dear Reviewer bFpj,
> > >
> > > Thank you for your constructive feedback and suggestions. We sincerely appreciate your positive assessment of our work's contribution. We will actively explore the use of larger-scale datasets during pretraining in our future research to enhance the framework's robustness. We are grateful for your insightful suggestions to strengthen our methodology.
> > >
> > > Best regards,
> > >
> > > The Authors

---

### Official Review · Reviewer_5obH · 2025-03-11

**Overall Recommendation:** 4

**Summary:**

This paper presents LangTime, a novel language-guided model for time series forecasting that addresses key challenges in leveraging large language models for this task. Specifically, the authors construct Temporal Comprehension Prompts to help LLMs understand domain-specific time series data, along with a new reinforcement learning-based fine-tuning algorithm, TimePPO, designed to mitigate error accumulation in autoregressive models. Extensive experiments on seven time series datasets demonstrate that LangTime achieves state-of-the-art forecasting performance, surpassing previous methods in both cross-domain generalization and autoregressive forecasting stability.

## update after rebuttal

I support for accept

**Claims And Evidence:**

The claims made in the paper regarding the novel application of LLMs to time series forecasting are well-supported by the presented evidence. In addition, the detailed experimental setup and the availability of the source code of the proposed model enhance the study’s reproducibility.

**Essential References Not Discussed:**

The authors have adequately discussed relevant works on large language models for time series forecasting and reinforcement learning within the context of large language models.

**Experimental Designs Or Analyses:**

The experimental design is comprehensive and well-executed. However, there are several issues that need to be addressed:
1. The paper introduces TimePPO and provides valuable results; however, the authors need to clarify the choice and tuning of the hyperparameters, such as \gamma, \xi in equation (7) and r(\theta), \eta in equation (8). More detailed explanations are needed regarding how these parameters are selected and how they impact model performance.
2. While the authors have demonstrated LangTime’s performance on a number of datasets, why did they not attempt to train LangTime using other large language models to assess its generalizability and adaptability?

**Methods And Evaluation Criteria:**

The methods presented in the paper are robust and well-suited for time series forecasting tasks, effectively addressing key challenges such as cross-domain generalization, cross-modality alignment, and error accumulation. In addition, the benchmark datasets are carefully selected to cover a wide range of time series characteristics, and the evaluation criteria are well-defined, allowing for a thorough assessment of the model's performance.

**Other Comments Or Suggestions:**

The authors provide informative figures, but additional explanatory captions or annotations are needed. For example, in Figure 6 (T-SNE visualization), a clearer distinction between the domains being compared and an explanation of why this visualization is important for understanding the model’s capabilities would be helpful. It is recommended that the authors review the manuscript and make the necessary adjustments.

**Other Strengths And Weaknesses:**

Strengths:
1. The introduction of Temporal Comprehension Prompts to guide large language models in understanding domain-specific time series data is a novel contribution. It effectively enhances the model's ability to interpret and process time series information.
2. The proposed TimePPO algorithm, designed to improve autoregressive forecasting by mitigating error accumulation, is an innovative approach. It addresses key challenges in long-term forecasting, ensuring more stable and accurate predictions over extended horizons.

Weaknesses:
1. While the model shows impressive forecasting accuracy, the complexity of its architecture and training process, particularly with the use of reinforcement learning, may raise concerns regarding scalability and efficiency. A discussion of its training time and computational requirements would be valuable.

**Questions For Authors:**

1. In Section 1 and Figure 1, the paper mentions cross-modality alignment. Could the authors elaborate on the difference between using language as prefixes and the proposed language-guided strategy for aligning modalities?
2. In Section 3.3, the authors adopt the previous token form <|EMB|> and <|OUT|> as the compressed token and prediction token. Could the authors explain why these specific token forms were chosen and what advantages they offer over alternative tokenization strategies?
3. The authors state that the mask rate is set to 0.4. Does this mask rate influence how the model learns and understands time series data? Specifically, how does the choice of mask rate impact the model's ability to extract temporal patterns and maintain prediction accuracy over longer horizons?
4. Does TimePPO fine-tune all parameters of the model, or are specific parts of the model frozen during fine-tuning?

**Relation To Broader Scientific Literature:**

LangTime situates itself within the growing body of work that applies large language models to time series forecasting. The authors provide a thorough discussion of related work, highlighting existing methods and clearly explaining how their approach advances the field by addressing the unique challenges of multi-domain generalization, cross-modality alignment, and error accumulation in autoregressive predictions.

**Theoretical Claims:**

The theoretical contributions of the paper are sound, well-supported, and free from any issues.

---

> ### Author Rebuttal · Authors · 2025-04-01
>
> We sincerely thank you for the thorough review and insightful comments. In our response, models were jointly pre-trained on the ETTh1 and Weather, and experiments involving the TimePPO stage were fine-tuned on individual datasets. We presented the average of the results, and full results are available at [link](https://anonymous.4open.science/r/full-E4EE/README.md).
>
> >E1: Hyperparameter Explanation
>
> For Eq. (7), we introduce parameter ξ based on GAE [1] to address error accumulation effects. When prediction steps increase, model errors often exceed those of repeating previous steps, causing underestimated advantages. The coefficient ξ(<1) relaxes this constraint to better reflect relative prediction quality. For the hyperparameters in Eq.(6) and Eq.(8), $\beta$ controls the MSE penalty term, aiming to regulate the extent of policy updates from the reward score perspective. For $\eta$, we referenced the alignment tax design in InstructGPT, enhancing the model's prediction capability and training stability. Sensitivity analysis experiments showed no significant impact.
>
> Tab.1: sensitivity analysis for $\beta$
> |||PT|0|0.01|0.03|0.05|0.1|0.3|0.5
> |-|-|-|-|-|-|-|-|-|-
> |ETTh1|MSE|0.447|0.440|0.440|**0.438**|0.438|0.439|0.440|0.440
> ||MAE|0.435|0.432|0.432|**0.431**|0.432|0.433|0.433|0.433
>
> Tab.2: sensitivity analysis for $\eta$
> |||PT|0|0.1|0.5|0.7
> |-|-|-|-|-|-|-
> |ETTh1|MSE|0.447|0.439|0.440|**0.438**|0.440
> ||MAE|0.435|0.434|0.433|**0.431**|0.432
>
> A detailed parameter discussion will be added in the paper's next version.
>
> [1] Schulman, J. et al. High-dimensional continuous control using generalized advantage estimation. ICLR2016.
>
> >E2: Performance Across Backbones
>
> The rationale for selecting Qwen and experimental results with other LLMs are detailed in **Q1** of our response to Reviewer **bFpj**, demonstrating adaptability across LLMs and GPT2's limitations.
>
> >W1: Training Efficiency Discussion
>
> For computational efficiency concerns, please refer to our response **W1** to Reviewer **bFpj**.
>
> >Q1: Difference Between Language Prefixes and Language-Guided Strategy
>
> Using language as prefixes inadequately describes the cross-modal relationship between language and time series, hindering LLMs' comprehension of unseen time series data during pre-training and causing modality misalignment. LangTime provides equivalent domain information while guiding LLMs through task-specific instructions (compression and prediction) and dual training objectives (reconstruction and prediction) to achieve alignment. For additional details, see our response to **Q2** from Reviewer **BdAz**. To validate LangTime's effectiveness in enhancing LLMs' time series understanding, we replaced the LLM with a linear layer. As shown in Tab.3, the significant performance decline confirms the necessity of our language-guided strategy.
>
> Tab.3: Impact of Removing LLM
> |backbones|Qwen||Linear||
> |-|-|-|-|-
> ||MSE|MAE|MSE|MAE
> |ETTh1|**0.448**|**0.431**|0.723|0.588
> |Weather|**0.270**|**0.277**|0.301|0.333
>
> >Q2:Could the authors explain why these specific token forms were chosen and what advantages they offer over alternative tokenization strategies?
>
> The special tokens <|EMB|> and <|OUT|> are newly added learnable tokens without predefined meanings for LLMs. Their specific forms were chosen primarily for human readability rather than offering inherent advantages in modality alignment. To verify this, we conducted experiments with alternative tokens (<|ABC|> and <|DEF|>), which showed no significant performance differences(Tab.4).
>
> Tab.4: Impact of Token Replacement
> |ETTh1|MSE|MAE
> |-|-|-
> |Original|0.448|0.431
> |Modified|0.448|0.432
>
> >Q3: How does the choice of mask rate impact the model's ability to extract temporal patterns and maintain prediction accuracy over longer horizons?
>
> The mask rate balances two objectives: (1) preventing overfitting in datasets with varying convergence speeds, and (2) enhancing temporal pattern learning through reconstruction tasks. Experiments show that extreme mask rates (too low/high) degrade performance: low rates fail to improve temporal pattern extraction, while high rates impair long-horizon predictions. Shorter prediction lengths suffer more from error accumulation(Tab.5).
>
> Tab.5: Impact of Mask Rate and Prediction Length on ETTh1
> |Length|24||96||192||
> |-|-|-|-|-|-|-
> |Mask Rate|MSE|MAE|MSE|MAE|MSE|MAE
> |0|0.466|0.442|0.444|0.431|0.460|0.437
> |0.2|0.451|0.438|0.441|0.432|0.453|0.433
> |0.4|0.455|0.435|0.440|0.430|0.458|0.435
> |0.6|0.461|0.438|0.444|0.432|0.451|0.435
>
> >Q4: Does TimePPO fine-tune all parameters of the model, or are specific parts of the model frozen during fine-tuning?
>
> In our experiments, TimePPO fine-tunes full parameters. However, freezing certain components still achieves similar performance, demonstrating the method's scalability.
>
> Tab.6: Effect of Fine-tuning Parameters
> |Fine-tuned Parameters|MSE|MAE
> |-|-|-|
> |pre-trained|0.439|0.432
> |Full|**0.435**|**0.431**
> |LLM(>97%)|**0.435**|**0.431**
> |TE(<3%)|0.437|0.432

---

> > ### Comment · Reviewer_5obH · 2025-04-03
> >
> > The authors present a well-motivated and methodologically sound contribution to the emerging area of LLM-based time series forecasting. They have clearly addressed the concerns raised during the review process and provided strong justifications for their design choices. The proposed LangTime framework effectively tackles key challenges and demonstrates strong performance across diverse benchmarks. I support acceptance.

---

> > > ### Author Response · Authors · 2025-04-04
> > >
> > > Dear Reviewer 5obH,
> > >
> > > Thank you for reviewing our manuscript and providing constructive feedback. We sincerely appreciate your recognition of our work's motivation and methodology, as well as your insightful comments that strengthened the research's rigor. We will carefully incorporate your suggestions to further refine the manuscript's technical depth and clarity.
> > >
> > > Best regards,
> > >
> > > The Authors

---

### Official Review · Reviewer_BdAz · 2025-03-13

**Overall Recommendation:** 2

**Summary:**

This paper introduces LangTime, a unified framework that leverages large language models (LLMs) for time series forecasting across multiple domains and modalities. The authors identify key challenges when applying LLMs to temporal data—Cross-domain generalization, cross-modality alignment and error accumulation in autoregressive frameworks. To address these, LangTime integrates Temporal Comprehension Prompts (TCPs), which serve as structured inputs to guide LLMs in interpreting time series data by embedding dataset-specific and variable-level context into a compact representation. Furthermore, the paper presents TimePPO, a reinforcement learning-based fine-tuning strategy tailored for time series, which introduces a multi-dimensional reward mechanism and a repeat-based value estimation to improve long-horizon prediction robustness. Through comprehensive empirical evaluation, LangTime demonstrates superior forecasting accuracy and strong generalization to previously unseen datasets. The study concludes by suggesting that future developments will explore extending LangTime’s capabilities to broader time series analysis applications.

**Claims And Evidence:**

The paper lacks a detailed explanation of how the dataset-wise and channel-wise information used in the Temporal Comprehension Prompts (TCPs) is generated. Additionally, it is unclear whether different formulations or descriptions of the data might affect the forecasting performance of LangTime. If the model is sensitive to these variations, how can its performance and generalizability be consistently guaranteed?

**Essential References Not Discussed:**

See above.

**Experimental Designs Or Analyses:**

I have checked the soundness of experimental designs. Deisgn of reconstruction loss can well enhance the alignment between time series and context information.

**Methods And Evaluation Criteria:**

Yes. The methods and evaluation criteria are appropriate.

**Other Comments Or Suggestions:**

See above.

**Other Strengths And Weaknesses:**

Strengths:

LangTime demonstrates state-of-the-art performance on established benchmark datasets and shows strong zero-shot generalization capabilities across previously unseen domains.

Weaknesses:

1）The novelty of the proposed method is limited, as similar concepts and methodologies have been presented in recent works such as TimeLLM, UniTime and MetaTST.

2）The paper does not provide a clear explanation of the methodology used for constructing the prompts. And it also lacks an evaluation of the model's generalizability under slight variations in the language prompt, which is important for assessing its robustness in real-world applications.

**Questions For Authors:**

1) Could you provide a detailed explanation of the prompt construction process in your approach, and further discuss the model’s generalizability when faced with slight variations in the prompt formulations?

2) In terms of aligning language with time series data, what are the key innovations of your method compared to existing approaches such as TimeLLM, UniTime, and MetaTST?

**Relation To Broader Scientific Literature:**

In fact, a growing body of work has explored the integration of language models with time series forecasting, including approaches such as MetaTST, TimeLLM, UniTime, and TimeMMD. It would be beneficial for the paper to further clarify its distinctions and advancements over these existing methods.

**Theoretical Claims:**

I have checked the correctness of theoretical claims.

---

> ### Author Rebuttal · Authors · 2025-04-01
>
> We sincerely thank you for the thorough review and insightful comments. In our response, our model was jointly pre-trained on the ETTh1 and Weather datasets, and experiments involving the TimePPO stage were fine-tuned on individual datasets. We presented the average of the results, and detailed experimental results are available at https://anonymous.4open.science/r/full-E4EE/README.md.
>
> >Q1:Could you provide a detailed explanation of the prompt construction process in your approach, and further discuss the model’s generalizability when faced with slight variations in the prompt formulations?
>
> The prompts in our approach consist of two parts. The first part includes common domain descriptive information used in existing methods to provide richer linguistic information. The second part has two task instructions to guide the model in understanding the time series data and generating predictions. We compared the impact on performance when these two parts varied separately, as shown in Table 1. However, when their expression changes but the basic meaning remains consistent, modifying the domain description and instructions does not significantly affect the model's performance.
>
> Table 1: Impact of language prompt modification on model capability
> | ETTh1 | MSE | MAE |
> |---|---|---|
> |Original| 0.448  | 0.431  |
> |Instruction Prompt| 0.449  | 0.433  |
> |Data Description Prompt| 0.450  | 0.434  |
>
> In Table 1, `Instruction Prompt` means changing TCP to the following format:
>
> ```
> The details of the provided time series:
> Period: <Timestamp>,
> Dataset: <Dataset Information>,
> Channel: <Channel Information>,
> Value: <Time Series Representation>,
> Please summarize this series in a single term: <|EMB|>.
> Using the given details, forecast the upcoming <N> values: <|OUT|>
> ```
> `Data description Prompt` means modifying the descriptive information of the following two datasets:
>
> ```json
> ETTh1: {
> Original: "An hourly-sampled electricity transformer dataset intended for electrical asset monitoring, collected from one area in a province in China."
> Modified: "An hourly-sampled dataset of electricity transformers designed for monitoring electrical assets."
> },
> Weather: {
> Original: "Meteorological indicator data with ten minute sample rate."
> Modified: "Data on meteorological indicators sampled every ten minutes."
> }
> ```
>
> >**Q2:** In terms of aligning language with time series data, what are the key innovations of your method compared to existing approaches such as TimeLLM, UniTime, and MetaTST?
>
> 1. **Characteristics of Existing Methods:** Existing approaches like TimeLLM and UniTime integrate linguistic information with time series data merely by concatenation, failing to emphasize **the relationship between the two parts**. This can make it challenging for LLMs, which have not encountered time series data during pre-training, to understand sequential correlations. MetaTST, on the other hand, does not employ LLMs as its backbone architecture, therefore it does not leverage LLMs' comprehension capabilities for time series data. While these methods leverage the comprehensive pre-trained knowledge of LLMs by providing linguistic information, they overlook the powerful instruction-following abilities of LLMs.
>
> 2. **Innovations of Our Method:** Our approach introduces diverse domain information via TCP and utilizes two directives to present LLMs with dual tasks: first, compress the presented information to generate a model understanding; then predict future outcomes based on this understanding (compressed token) following instructions. Benefiting from LLMs' robust instruction-following capacity, our language-guided approach aids LLMs in comprehending time series data by integrating reconstruction and prediction training objectives to fully exploit LLM's time series understanding and prediction abilities. Additionally, LangTime employs an autoregressive structure, allowing flexible prediction length. Inspired by RLHF and considering time series data characteristics, we designed a reward function with multi-angle assessment and a repetition strategy-based Value function. Fine-tuned via PPO algorithm, this enhances the model's ability to combat cumulative errors.
>
> To validate our method's effectiveness in enhancing LLMs' time series understanding, we replaced the LLM with a linear layer. As shown in Table 2, the significant performance decline confirms the necessity of our language-guided strategy.
>
> Table 2: Impact of Removing LLM
> |backbones|Qwen||Linear||
> |-|-|-|-|-
> ||MSE|MAE|MSE|MAE
> |ETTh1|**0.448**|**0.431**|0.723|0.588
> |Weather|**0.270**|**0.277**|0.301|0.333

---

### Official Review · Reviewer_8SdX · 2025-03-13

**Overall Recommendation:** 3

**Summary:**

The paper introduces LangTime, an approach that builds on top of existing large language models (LLMs) to effectively perform time series forecasting. The paper identifies 3 crucial problems with adapting LLMs for forecasting tasks - cross-domain generalization, cross-modality alignment, and error accumulation in autoregressive frameworks.

LangTime uses a temporal encoder to convert non-overlapping patches of the time series into tokens. These tokens are provided as inputs to the LLM, and the model is additionally trained to reconstruct the time series from the tokens apart from forecasting. For cross-modality alignment, the paper proposes using Temporal Comprehension Prompts (TCP) to provide additional information about the time series domain and specific channel characteristics. To reduce error accumulation, the paper introduces a novel training procedure using Proximal Policy Optimization (TimePPO), similar to RLHF in Natural Language Processing.

## update after rebuttal
The authors have addressed my concerns and I am satisfied with the response. Accordingly, I have increased the score from 2 to 3.

**Claims And Evidence:**

Using TCP (guidance through detailed textual description) helps in improving the cross-modal alignment, as witnessed in Table 3. However, similar approaches are used in TimeLLM and UniTime with respect to using text prompts to describe the domain of the time series. Additionally, UniTime also employs similar reconstruction losses. In my opinion, a comparison of LangTime SFT (supervised fine tuning) results with UniTime SFT and TimeLLM to show quantitative improvements provided by TCP will be useful. The comparison is available in Tables 1 and 2, but the results look mixed with LangTime SFT underperforming in multiple cases.

Similarly, the effect of TimePPO towards minimizing the error accumulation is not clearly explained. For example, if $\eta$ is a relatively large value, then isn't the loss essentially the same as in the SFT case? Also, does the predicted output sequence length affect the error accumulation? This has been shown in recent time series foundation models like TimesFM.

**Essential References Not Discussed:**

The paper covers multiple SOTA prior works that are related to adapting LLMs for forecasting.

**Experimental Designs Or Analyses:**

Yes, I have checked the soundness/validity of experimental designs. Tables 2,3, and 4 showcase the ablation with respect to loss functions (SFT vs PPO), language guidance, and reward functions for PPO, respectively. However, a few more ablations are required to highlight the different contributions in the paper. For example, what are the effects of $\beta$ and $\eta$ in equations 5 and 8, respectively?

**Methods And Evaluation Criteria:**

Yes, the paper tests the proposed approach on standard time series forecasting datasets. Additionally, the chosen baselines are valid.

**Other Comments Or Suggestions:**

The TimePPO section (Section 3.4) requires more clarity. Some coefficients ($\beta$, $\gamma$, $\lambda$, $\tau$, $\eta$) are not clearly explained.

In Algorithm 1, in line 6, if the objective in Eq 8 has to be maximized, doesn't that imply maximizing $\|y - \hat{y}\|$?

Eq 6 is flowing out of the column width.

**Other Strengths And Weaknesses:**

Strengths:
1. The paper provides a set of extensive experimental results that showcase the effectiveness of the proposed approach.
2. The provided ablations are useful in understanding the overall approach.
3. The qualitative analyses through t-SNE and attention maps provide more interpretability to the proposed approach.

For Weaknesses, please check the "Claims and Evidences" and "Experimental Designs" sections.
One of the main weaknesses is the delineation of the 3 contributions. The paper highlights cross-domain generalization and cross-modal alignment as challenges, but it is unclear which components address the cross-domain generalization part. The temporal encoders and reconstruction loss help in cross-modal alignment, and the TimePPO objective helps with better forecasting. Also, since the overall setup is trained on a per-dataset basis, why is there a need for cross-domain generalization? Additionally, the experimental results do not showcase cross-domain generalization.

Additionally, I think the paper might benefit from zero-shot comparisons against other time series foundation models like TimesFM, Chronos, etc.

**Questions For Authors:**

1. How is cross-domain generalization addressed/achieved in this paper?
2. How does the TCP used in this paper differ from the language prompts used in UniTime and TimeLLM?
3. What are the effects of $\beta$ and $\eta$ in equations 5 and 8, respectively?
4. What is the performance comparison of the pre-trained model against zero-shot time series foundation models?
5. Why does TimePPO specifically contribute towards reducing error accumulation? Is there any experimental evidence to support this claim?
6. Can the authors describe the output sequence length used in the paper, and how does that affect the error accumulation?

If the authors can respond to 1,2,5, and 6, I am willing to raise the score.

**Relation To Broader Scientific Literature:**

Leveraging LLMs trained on large-scale internet textual data effectively for forecasting is a key problem. The effective adaptation of LLMs, through the approaches shown in the paper, can unlock the ability to obtain accurate forecasts with limited fine tuning. Additionally, this allows for future work in the direction of reasoning about the generated forecasts in natural language.

Specifically, from my domain knowledge in forecasting, the application of PPO to improve forecasts is novel and interesting. And from the experimental results, such finetuning shows promising results.

**Theoretical Claims:**

There are no theoretical claims in the paper.

---

> ### Author Rebuttal · Authors · 2025-04-01
>
> As for Eq.(8), it should actually be: $-||y-\hat{y}||^2_2$ (refer to `actor_loss_fn` in `ppo_trainer.py`). We will modify the description of hyperparameters and Eq.(6) and Eq.(8) in the next version of the paper. Full results are available https://anonymous.4open.science/r/full-E4EE/README.md.
> > C1: Comparison of LangTime SFT with UniTime SFT
>
> Unlike UniTime and TimeLLM, LangTime uses a more flexible autoregressive structure. SFT focuses on the process of generating the next prediction, which has limited effectiveness on the continuous generation process of autoregressive models. This explains why LangTime performs poorly during SFT.
> > Q1: Cross-Domain Generalization
>
> We need to clarify that LangTime undergoes joint **pre-training on 7 datasets**, which enables it to understand time series across different domains. When facing unseen domains, we provide description information for each channel in the new dataset via TCP. Leveraging the rich language knowledge of LLM and the understanding of time series across different domains acquired during pre-training, LangTime addresses the cross-domain generalization issue. We demonstrate its generalization capability in Table 5 of the paper. We conducted pre-training on ETTh1 and ETTm1, and evaluated on 3 other datasets.
>
> Tab.1: Cross-domain Generalization Ability
> ||LangTime||UniTime||
> |-|-|-|-|-
> ||MSE|MAE|MSE|MAE|
> |ETTm2|**0.301**|**0.335**|0.306|0.343
> |Weather|**0.320**|0.335|0.323|**0.334**
> |ECL|**0.377**|**0.442**|0.458|0.529
>
> >Q2: How does the TCP used in this paper differ from the language prompts used in UniTime and TimeLLM?
>
> The prompt-as-prefix approach (UniTime/TimeLLM) cannot fully describe **the connection between the language part and the time series part**, making it difficult for the LLM to understand time series data that it has never seen during the pre-training, leading to challenges in modality alignment. LangTime provides the same domain information as the prompt-as-prefix approach while guiding the LLM to **compress and predict time series through two task instructions**. Modality alignment is achieved through reconstruction and prediction tasks. You can also refer to our response to reviewer **BdAz**'s **Q2** for more details.
> > Q3: Effects of $\beta$ and $\eta$
>
> Due to length constraints, please refer to our response to reviewer **5obH**'s **E1**.
>
> >Q4: Comparison of LangTime against foundation model
>
> Because TimesFM is pre-trained on a large number of datasets, to maintain the invisibility of the test domain, we re-trained LangTime on the Weather, ECL, and Exchange datasets, and conducted zero-shot testing on the ETT dataset. As shown in Tab.2, LangTime achieved better performance on most datasets, demonstrating its effectiveness in cross-domain generalization.
>
> Tab.2: Zero-shot performance comparison of LangTime and foundation model
> ||LangTime||TimesFM||
> |-|-|-|-|-
> ||MSE|MAE|MSE|MAE
> |h1|**0.537**|**0.481**|0.671|0.502
> |h2|**0.416**|**0.429**|0.471|0.436
> |m1|0.907|0.615|**0.789**|**0.561**
> |m2|**0.316**|**0.358**|0.422|0.386
>
> >Q5: Why does TimePPO specifically contribute towards reducing error accumulation?
>
> Autoregressive models have flexible prediction lengths but are significantly affected by accumulated errors because they lack the ability to discern whether the output of the previous step is reliable. TimePPO estimates the return for the entire sequence through the Value Function and evaluates the long-term value of the current step relative to the estimated level using advantages. These designs optimize the prediction of the entire sequence rather than just focusing on the model's ability to predict the next step. Thus, TimePPO helps alleviate the cumulative error issue in autoregressive models. To verify this, we compared the model's metrics of the last step in long-term predictions (336, 720). As shown in Tab.3, TimePPO performs better in the final step predictions most affected by accumulated errors.
>
> Tab.3: TimePPO'role in error accumulation
> ||ETTh1||Weather||
> |-|-|-|-|-
> ||MSE|MAE|MSE|MAE
> |PT|0.534|0.491|0.415|0.385
> |SFT|0.533|0.490|0.419|0.386
> |TimePPO|**0.528**|**0.489**|**0.410**|**0.381**
>
> >Q6: How does the output sequence length affect the error accumulation?
>
> The output sequence length used in this paper is 96 (the prediction length for each step in autoregression). Since errors occur during each prediction, they accumulate continuously during iterations. If the single output sequence length is small, more steps are needed to predict the same length, making accumulation errors more pronounced and prediction performance relatively poorer. Conversely, although reducing the number of steps, predicting a longer sequence in a single prediction may lead to worse results. To verify this, we compared the impact of different single prediction lengths on final performance.
>
> Tab.4: Impact of single prediction length
> |ETTh1|24|48|96|144|192
> |-|-|-|-|-|-
> |MSE|0.461|**0.448**|**0.448**|0.451|0.458
> |MAE|0.438|0.432|**0.431**|0.434|0.435

---

> > ### Comment · Reviewer_8SdX · 2025-04-06
> >
> > The authors have addressed my concerns and I am satisfied with the response. Accordingly, I will increase the score.

---

> > > ### Author Response · Authors · 2025-04-06
> > >
> > > Dear Reviewer 8SdX,
> > >
> > > Thank you for taking the time to review our rebuttal and adjust the score. We sincerely appreciate your constructive feedback, which has greatly helped improve our work. Your expertise and thoughtful evaluation are invaluable to us. Thanks again for your time and consideration.
> > >
> > > Best regards,
> > >
> > > The Authors

---

### Decision · Program_Chairs · 2025-05-01

**Decision:**

Accept (poster)

**Comment:**

This work proposes a new approach to time series forecasting that uses a pre-trained LLM backbone and addresses three challenges with recent unified time series models. They introduce a training objective based on RLHF and show that it improves their model's zero-shot and fine-tuned performance on standard datasets.

The reviewers lean positively, noting many strengths, especially in thorough experiments with ablations and qualitative analyses and in introducing TimePPO to train forecasting models and TCPs to aid in domain adaptation. The weaknesses discussed mainly include some clarifications that should be incorporated into the camera-ready paper and the datasets are also quite over-studied in the literature at this point. Comparisons with other state-of-the-art methods like MOMENT and UniTS would help and further ablations on the need for the pre-trained LLM vs. simpler architectures would also further strengthen this work.

Overall, the strengths identified by the reviewers outweigh the negatives and I suggest accepting this paper.